# Patterns of call communication between group-housed zebra finches change during the breeding cycle

**Lisa F Gill\*, Wolfgang Goymann, Andries Ter Maat, Manfred Gahr**

Max Planck Institute for Ornithology, Seewiesen, Germany

**Abstract** Vocal signals such as calls play a crucial role for survival and successful reproduction, especially in group-living animals. However, call interactions and call dynamics within groups remain largely unexplored because their relation to relevant contexts or life-history stages could not be studied with individual-level resolution. Using on-bird microphone transmitters, we recorded the vocalisations of individual zebra finches (*Taeniopygia guttata*) behaving freely in social groups, while females and males previously unknown to each other passed through different stages of the breeding cycle. As birds formed pairs and shifted their reproductive status, their call repertoire composition changed. The recordings revealed that calls occurred non-randomly in fine-tuned vocal interactions and decreased within groups while pair-specific patterns emerged. Call-type combinations of vocal interactions changed within pairs and were associated with successful egg-laying, highlighting a potential fitness relevance of calling dynamics in communication systems.

## Introduction

Vocal communication plays an important role for a variety of social animals, because it is often directly linked with individual survival and successful reproduction. Vocal signals are especially important in group-living species, because they can be used to maintain group cohesion and coordinate common activities (reviewed in *Fichtel and Manser, 2010*), but also to recognise, locate and interact with specific individuals inside such groups (*Jouventin et al., 1999a*, *1999b*; *Aubin and Jouventin, 2002*; *Balsby et al., 2012*; *Ter Maat et al., 2014*). However, in songbirds, most vocalisation studies have focused on male song and its relationship with hormones and reproduction (*Nottebohm et al., 1987*; *Perez et al., 2012*; *Gahr, 2014*) in solitary, territorial, temperate-zone species (*Marler, 2004*). But songbirds also produce calls in a variety of contexts (*Marler, 2004*), sometimes in very large numbers throughout the day (*Beckers and Gahr, 2010*). The usage and function of such calls is still unknown, mainly because it has been challenging to investigate with individual-level resolution naturally occurring vocal interactions between group members, in relevant contexts or different life-history stages.

It has been hypothesized that zebra finches (*Taeniopygia guttata*), which are group-living, socially monogamous and opportunistically breeding songbirds (*Zann, 1996*; *Perfito et al., 2007*), share a different form of vocal communication with their life-long partner compared to other members of their group (*Zann, 1996*). In this species, both sexes produce diverse calls in large numbers, in various social contexts (*Zann, 1996*; *Beckers and Gahr, 2010*; *Ter Maat et al., 2014*), and depending on group structure (*Elie et al., 2011*). Some soft calls, that is, low-amplitude vocalisations used in close-range signalling (*Dabelsteen et al., 1998*), have been suggested to play a role in pair communication (*Zann, 1996*; *Elie et al., 2010*; *Ter Maat et al., 2014*). However, until recently, it was not possible to record and reliably assign all calls of individual zebra finches in the presence of their mates and within a group (*Elie et al., 2011*). Because calls are short and may be low in amplitude, especially when used at close range, earlier studies often resorted to strongly reduced social contexts or

\*For correspondence: lgill@orn.
mpg.de

**Competing interests:** The authors declare that no competing interests exist.

**Reviewing editor**: Ian T Baldwin, Max-Planck Institute for Chemical Ecology, Germany

**eLife digest** As the name implies, songbirds produce song, but they may also emit large numbers of shorter calls. Because calls are often given in social situations, they are difficult to record and to assign to the correct individual. Therefore, it is still unclear what information is communicated by these calls and how important they are.

Zebra finches are highly vocal songbirds, with males singing and both females and males producing calls. In their natural habitat, Australia, these chatty birds pair for life and live in groups. To ensure successful breeding, zebra finches need to begin breeding activities as soon as the unpredictable environment allows. Therefore, even in captivity, they will readily breed when given nesting material.

To find out about the role of zebra finch calls in relation to different environmental or social factors, Gill et al. brought together in groups female and male zebra finches that had not met before, and followed their individual calls during different breeding stages. This was done using a technique called microphone telemetry that involves placing tiny wireless microphones on the birds.

The finches quickly formed breeding pairs, and when provided with nesting material, began building nests and laying eggs. While doing so, and especially when pairs began building nests, the birds changed how often they used certain calls and started using different call types; for example, they made more breeding-related 'cackles'. Calls often featured precisely timed back-and-forth calling interactions, and, over time, were directed more and more towards their partner than other members of the group. Pairs that performed more of these call exchanges during nesting were more likely than others to lay a clutch of eggs.

Overall, Gill et al. show that both the timing and types of calls used in pair communication are important for successful breeding. Future research could investigate the role of calls in group communication in more detail—possibly even in the wild—and how calling behaviour is reflected in the brain.

impoverished environments to investigate vocalisations at the individual level (*Blaich et al., 1996*; *Vignal et al., 2004*; *Anisimov et al., 2014*). Hence, these studies mainly addressed mechanistic questions of vocal behaviour, but to understand the underlying meaning of calling, individual-based information in a socially relevant context is necessary.

Our aim was thus to investigate with individual-level resolution the calling behaviour of zebra finches that were behaving freely in social groups, in a changing environment. We aimed at studying the dynamics of different call types and their usage—on the individual level as well as in interactions between mates and other group members—in relation to reproductive state and successful egg-laying. To do this, we housed groups of four females and four males previously unknown to each other together in large aviaries for about 3 weeks, provided them with nest material, and continuously recorded individual vocalisations of all group-members using microphone telemetry (= on-bird microphone transmitters). While the birds formed pairs and passed through different stages of their breeding cycle, we recorded vocalisations, performed behavioural observations, took blood samples for hormone determination, and monitored their nests to measure reproductive performance.

## Results

When housed together in social groups in large aviaries (*Figure 1—figure supplement 1*), female and male zebra finches that were equipped with individual on-bird microphones (*Figure 1*) (*Video 1*) and that were previously unknown to each other, formed pairs and started to breed after nesting material had been added (*Figure 2*). At the same time, the call repertoire (*Figure 3*) of individually recorded birds changed (*Figure 4*, *Figure 4—figure supplement 1*), and vocal interactions in the group became increasingly pair specific (*Figure 5*, *Figure 5—figure supplement 1*). In within-pair calling interactions, the combination of call types involved changed (*Figure 5*), and was related to successful egg-laying (*Figure 6*).

### Pair formation and reproductive stages

30 out of 32 birds formed pairs (for definitions, see 'Materials and methods', *Tables 1, 2*) within 1 to 7 days and began occupying nest boxes within 5.6 ± 3.3 days (mean ± standard deviation [SD]).

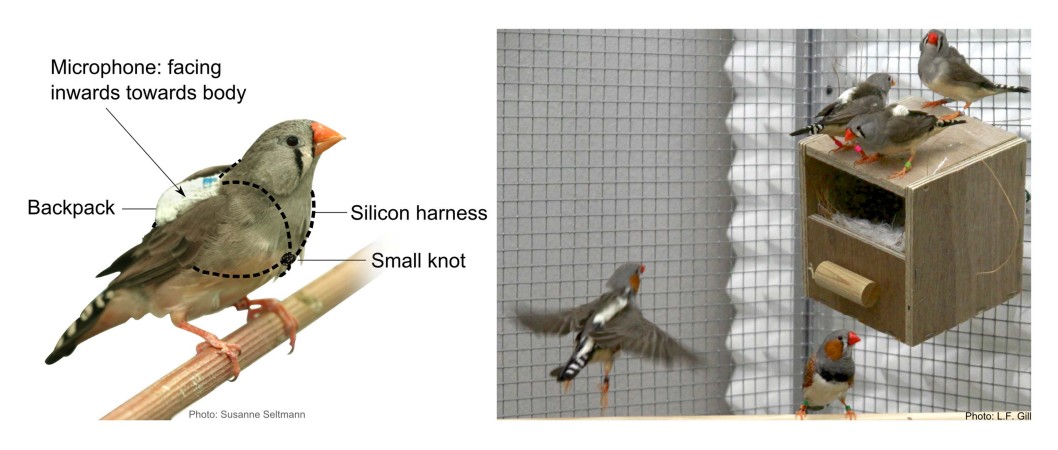

**Figure 1**. Position of on-bird microphones on freely behaving zebra finches. Close-up pictures of backpacks fitted on zebra finches in an aviary. The visible white 'backpack' contains a microphone (at the bottom), a radio transmitter, and a battery. It is placed on the bird's back (centre of mass) in a way that the microphone faces inwards, that is, towards the bird's body. Unlike the backpacks, harnesses disappear under the birds' feathers (left and right panel and *Video 1*), therefore, to demonstrate how they were fixed on the birds, the two silicon loops around the head and the abdomen are represented by dashed lines in the left panel. They were closed at the front of the bird with a small knot.
The following figure supplement is available for figure 1:

**Figure supplement 1**. Group housing.

The birds were not entirely synchronized in their reproductive stages and some nest building began even before actual nest material was provided (using scraps of food or single threads from backpack material). But the addition of proper nest material triggered nesting activities in all birds but one, resulting in a mean onset of nest building at 7.3 ± 2.4 days. Two categories of reproductive status, 'nest stage' and the more detailed 'breeding stage' resulted from our behavioural observations and nest checks (see 'Materials and methods') and are depicted in an overview of reproductive activity over the first 20 days (*Figure 2*). Hereby, 'nest stage' reflected large-scale changes of reproductive stages and was confirmed by correlated changes in sex steroid levels (*Figure 2—figure supplement 1*, see Appendix 1). Zebra finches reproduce as soon as environmental conditions permit to ensure successful breeding (see 'Discussion'). Therefore, birds that produced a clutch of eggs during the trials were termed 'successful' and those that did not as 'unsuccessful' at egg-laying. Pairs had eggs after 11.6 ± 5.8 days and began incubating them after 18.3 ± 5.1 days.

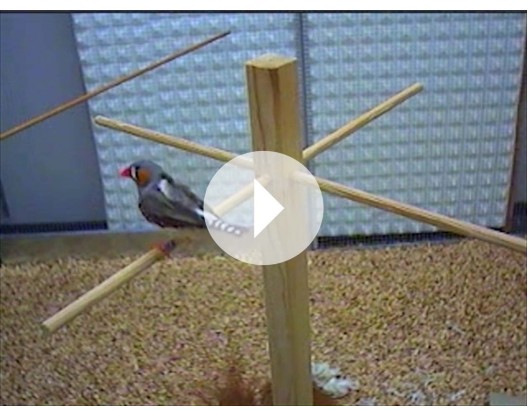

**Video 1.** Zebra finches behaving freely with on-bird microphones. Example video and external audio recording of Zebra finches behaving freely inside aviary (partial view) on the day of nest material. In the video, note the small white objects on the birds' backs ('microphone backpacks') that allow normal behaviours, for example, flight and collection of nest material. In the audio, note the soft, overlapping vocalisations, and wing beat sounds (see *Figure 7*).

## Vocalisation types and reproductive stages

Apart from calls related to parental behaviour ('thucks'), we found all vocalisation types described by *Zann (1996)*. The five most frequently

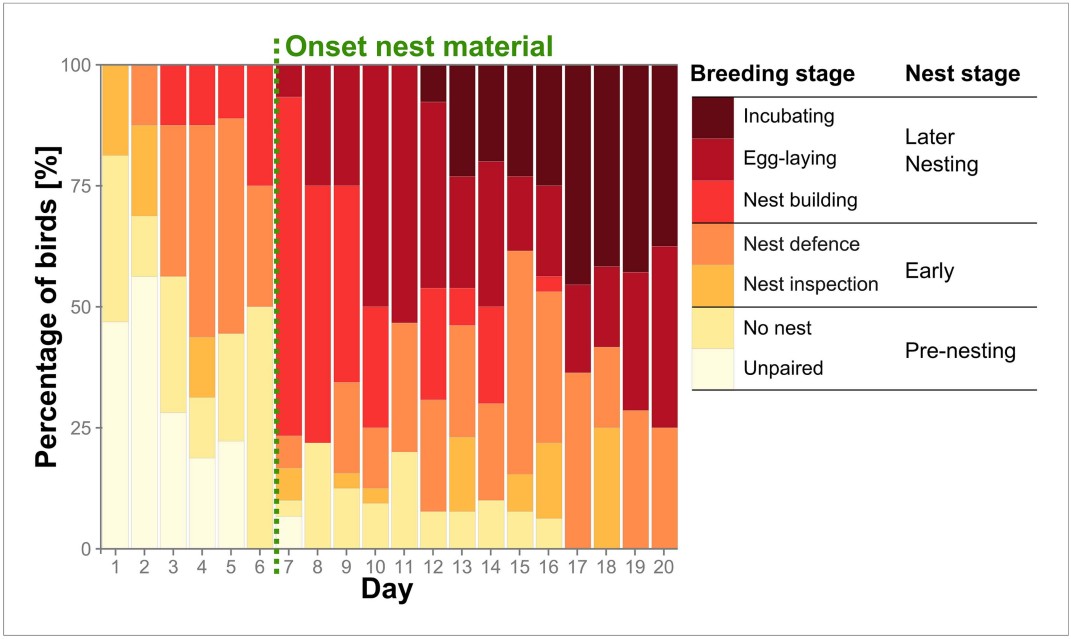

**Figure 2**. Group reproductive stages change over time. Percentage of birds ($n_{birds}$ = 32) assigned to the seven detailed breeding stages (coloured bars) and the three corresponding condensed nest stages over the first 20 days of the trials. Earliest onset of nest material provisioning (day 7) is indicated by a green dashed vertical line. Source data are available at http://dx.doi.org/10.5061/dryad.vt69s.

The following figure supplement is available for figure 2:

**Figure supplement 1**. Differences in steroid hormone concentrations at baseline levels and the three nest stages.

used and distinct call types ('distance calls', 'stacks', 'tets', 'cackles' and 'whines', *Figure 3*) were used for further analyses (182.752 calling events). Within these five call types, we found differences in the number of vocalisations uttered per bird and recording in relation to reproductive stage and sex (*Figure 4*, *Figure 4—figure supplement 1*, Appendix 2, *Appendix 2—table 1*, see 'Materials and methods' for sample sizes). For distance calls, cackles and whines, the numbers of vocalisations changed for males and females in the same way (no interaction between 'nest stage' and sex). Loud distance calls were produced most when birds were not yet paired or nesting, with highest levels during pre-nesting, that then decreased during the early and again during the later nest stages (*Figure 4*, $F_{stage}$ = 18.5, $F_{sex}$ = 1.13, $R^2_{marginal}$ = 0.20, $R^2_{conditional}$ = 0.37; see 'Materials and

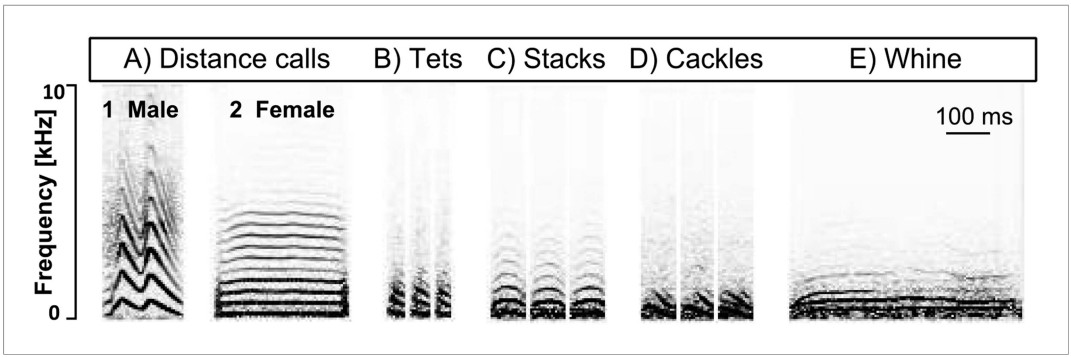

**Figure 3**. Call types used in our study. Example spectrograms of female (**A1**) and male (**A2**) distance, tet (**B**), stack (**C**), cackle (**D**), and whine (**E**) calls. x-axis: time [ms], y-axis: frequency [Hz].

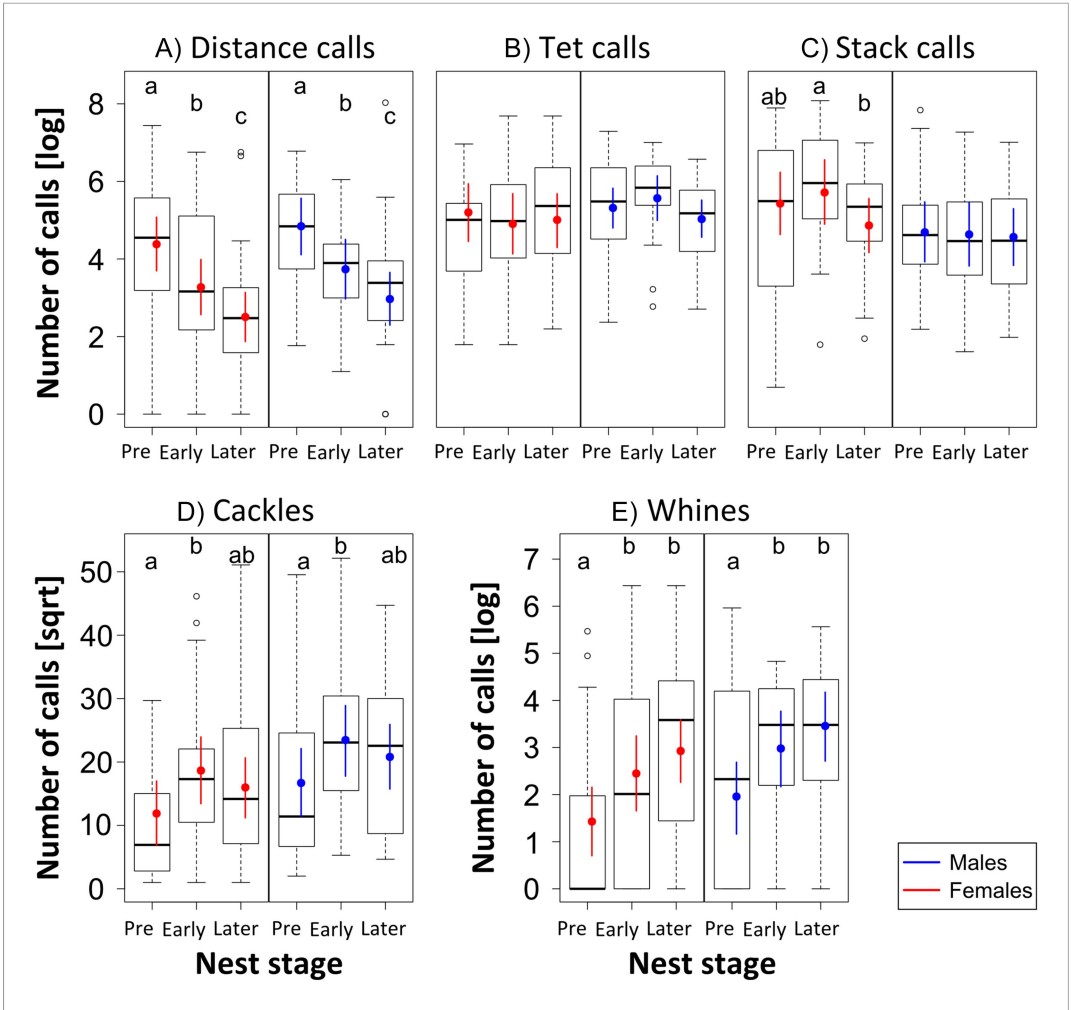

**Figure 4**. Female and male call-type usage at different nest stages. Boxplots of the number of vocalisations (natural log- or square-root transformed) per 4 hr of recordings for the different vocalisation types (**A**-**E**) in relation to the three Nest stages, analysed separately for females (red, $n_{females}$ = 12) and males (blue, $n_{males}$ = 10) from the three trials. They show that call types change differently over Nest stages: distance calls decrease (**A**), and cackles (**D**) and whines (**E**) increase. Thick black horizontal line = median of observations, box = 25–75% quantile of the observations (length = interquartile range), whiskers = last observation within 1.5 times the interquartile range from the edge of the box, circles = observations farther than 1.5 time the interquartile range from the edge of the box, coloured point = fitted value (Bayesian estimate), coloured vertical bar = 95% credible intervals (CrI) of the fitted value. If Bayesian estimates (coloured points) and CrI (vertical coloured lines) do not overlap inside single plots, there is a difference in the number of vocalisations used in relation to Nest stage. Such differences are indicated by different letters at the top of each box. Sample sizes during Pre-, Early and Later Nest stage were: 24, 20, and 35 data points coming from 9, 10, and 12 females, and 23, 18, and 26 data points coming from 8, 8, and 10 males. Source data are available at http://datadryad.org/review?doi=doi:10.5061/dryad.vt69s.

The following figure supplement is available for figure 4:

**Figure supplement 1**. Changes in call repertoire at more detailed breeding stages.

methods' for an explanation of F, $R^2_{marginal}$, $R^2_{conditional}$). Cackles and whines increased at the onset of reproductive activities, with cackles showing a peak in both sexes during the early nest stage ($F_{stage}$ = 3.80, $F_{sex}$ = 2.28, $R^2_{marginal}$ = 0.08, $R^2_{conditional}$ = 0.33; *Figure 4*), and whines showing a peak during early and later nest stages ($F_{stage}$ = 8.34, $F_{sex}$ = 1.58, $R^2_{marginal}$ = 0.12, $R^2_{conditional}$ = 0.24; *Figure 4*). Tets did not change throughout the three nest stages for either sex ($F_{stage}$ = 0.49, $F_{sex}$ = 0.39,

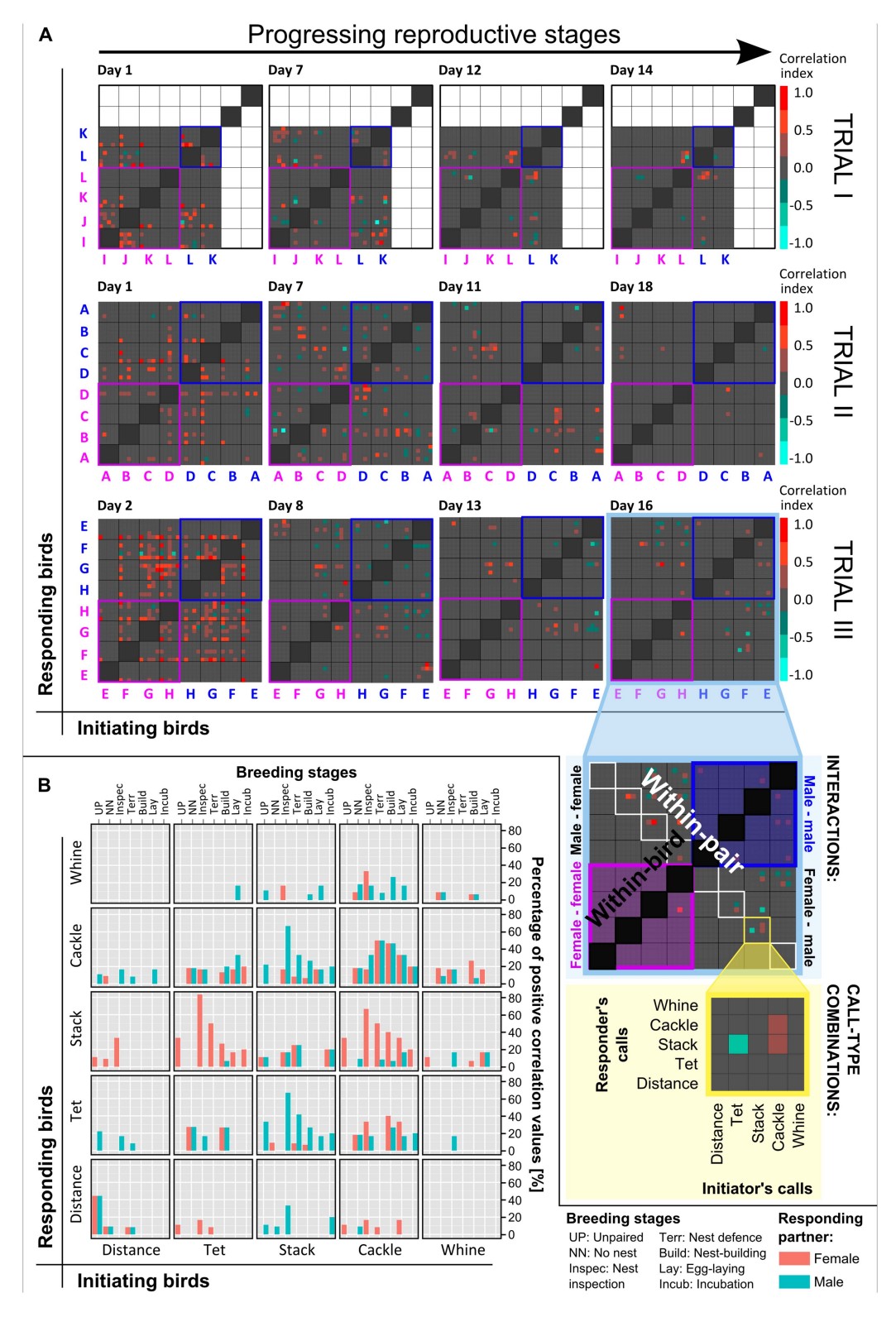

**Figure 5.** Vocal interactions within groups across reproductive stages. (**A**) Vocal interaction matrices. Examples of vocal correlation indices (from −1 to 1, see colour scale) resulting from PSTHs for all bird and call-type combinations during different phases of the trials (different days indicated above each box), for trials I, II, and III ($n_{birds}$ = 6, 8 and 8). All initiating birds (x-axis) and responding birds (y-axis) are represented by capital letters (pink: females, blue: males) and are subdivided into the five call types. Note that grey squares (= zero values) indicate there was no significant interaction in the respective dyad

*Figure 5. continued on next page*

*Figure 5. Continued*

and does not mean there were no vocalisations (see 'Materials and methods'). Same capital letters indicate members of a pair, and within-pair interactions can be found in the diagonal from top left to bottom right. Note an increase in within-pair interactions and a decrease in overall group interactions with progressing reproductive stages (left to right). Inserts in *Figure 5A* (lower right corner) explain the different interaction levels in the group (highlighted in pale blue) and the call-type interactions (highlighted in pale yellow). The dark grey diagonal from bottom left to top right represents within-bird interactions which were excluded from the analyses. Same-sex interactions are emphasized by pink (female–female) or blue (male–male) outer lines. In trial I, white squares represent missing values. The dataset is available at http://datadryad.org/review?doi=doi:10.5061/dryad.vt69s. (**B**) Within-pair vocal interactions at different breeding stages. Summary graph of positive within-pair calling interactions in relation to different call-type combinations, sex and the detailed breeding stages ($n_{pairs}$ = 10). Initiating birds' call types are plotted on the x-axis and percentages of positive responses (pink: females, blue: males) are plotted on the y-axis, in the corresponding call types. Note that both females and males were initiating and responding birds. Source data are available at http://datadryad.org/review?doi=doi:10.5061/dryad.vt69s.

The following figure supplement is available for figure 5:

**Figure supplement 1**. Increasing specificity of within-pair vocal interactions.

$R^2_{marginal}$ = 0.01, $R^2_{conditional}$ = 0.34; *Figure 4*). Stack calls showed an interaction between nest stage and sex: the amount did not change for males ($F_{stage}$ = 0.06, $R^2_{marginal}$ = 0.001, $R^2_{conditional}$ = 0.42; *Figure 4*), but in females, stacks were produced slightly more often during the early nest stage than during the pre-nesting stage ($F_{stage}$ = 2.37, $R^2_{marginal}$ = 0.05, $R^2_{conditional}$ = 0.31; *Figure 4*).

In sum, this shows that call-type usage, that is, the repertoire composition, changed at the individual level over the breeding cycle. Because the recordings were gained in temporal synchrony between all

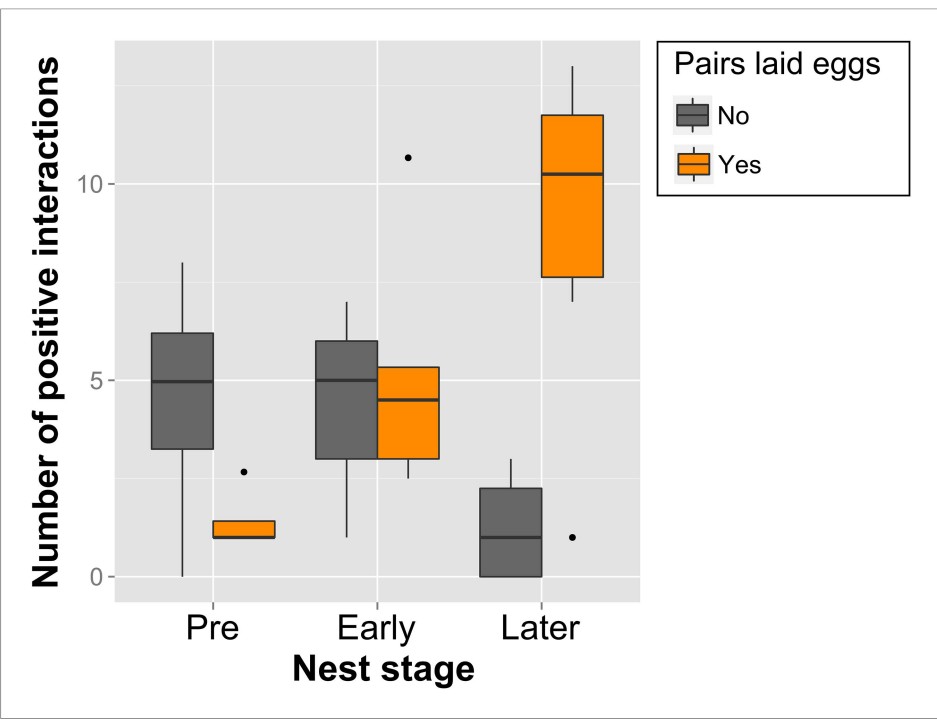

**Figure 6**. Call-type combinations associated with nest stages and successful egg-laying. Boxplot of within-pair number of combinations with significant interaction (positive) over Pre- ($n_{Pre}$ = 8), Early ($n_{Early}$ = 8) and Later Nesting ($n_{Later}$ = 10) for pairs that were successful (orange, n = 6) and unsuccessful (grey, n = 4) at producing a clutch of eggs within the 3-week trials (Later Nest stage here refers only to Nest-building). Note the increase in call-type interactions of successful pairs across the Nest stages. Thick black horizontal line = median of observations, box = 25–75% quantile of observations (length = interquartile range), whiskers = last observation within 1.5 times the interquartile range the edge of the box, black dots = observations farther than 1.5 time the interquartile range from the edge of the box. Source data are available at http://datadryad.org/review?doi=doi:10.5061/dryad.vt69s.

Table 1. Overview and short description of different agonistic, affiliative or sexual, and neutral behaviours and whether they were measured as occurrences (count) or every 2 min (duration)

| Behaviour | Description | Count/Duration | Type |
|---|---|---|---|
| Displacement | Focal bird arrives at another bird's location forcing it to leave | Count | Agonistic |
| Fighting | For example, bill-fight, full body fight, chasing | Count | |
| Clumping | Birds sit in direct physical contact with each other | Duration | Affiliative or sexual behaviour |
| Allopreening | One bird preens another bird | Duration | |
| Copulation solicitation | Female fans tail at male | Count | |
| Copulation | Male mounts female | Count | |
| Enter nest box | Birds enter the same nest box without fighting | Count | |
| Foraging | Bird is foraging on ground, feeding, drinking | Duration | Neutral |
| Preening | Bird is self-preening | Duration | |
| Flying | Bird flies around in aviary | Count | |
| Incubating | Bird sitting inside nest box with eggs | Duration | |

group members (see 'Materials and methods', *Figure 7*), our study also allowed investigating how individual birds used calls to interact with other individuals of the group.

## Vocal interactions, reproductive stages and successful egg-laying

These interaction-level data indicated that vocal networks were dynamic, differed in pair and group communication, and were related to the breeding stages. Peristimulus time histograms (PSTHs, for details see 'Materials and methods' and *Ter Maat et al., 2014*) that compare the onsets of the birds' vocalisations relative to each other revealed that calls did not occur randomly. Instead, in many cases, the calling behaviour of a specific bird elicited significant changes in the calling behaviour of another bird within a time frame of 0.5 s (relative to each dyad's baseline). The resulting correlation indices (see 'Materials and methods' for calculation and sample sizes) were plotted in confusion matrices showing all possible combinations of birds and call types (*Figure 5A*). These matrices demonstrated dynamic interaction patterns between the birds: while birds shared many 'significant interactions' with other birds in various call types at the beginning of the group trial, interactions decreased and became more and more specific with progressing reproductive activities (*Figure 5A*). On the day that nest material was provided, the diagonal between the top left and the bottom right lit up (*Figure 5A*),

Table 2. Overview and short description of breeding stages and nest stages

| Breeding stage | Description | Nest stage |
|---|---|---|
| Unpaired | Bird does not show increased prosocial behaviour towards specific individual | Pre-nesting |
| No nest | Paired but without nest | |
| Territorial | Pair defending nest site without nest material | Early nesting |
| Nest inspection | Pair inspecting different nest boxes | |
| Nest building | Pair bringing nest material to nest box | Later nesting |
| Laying | Pair's female laying eggs | |
| Incubation | Pair members incubating | |

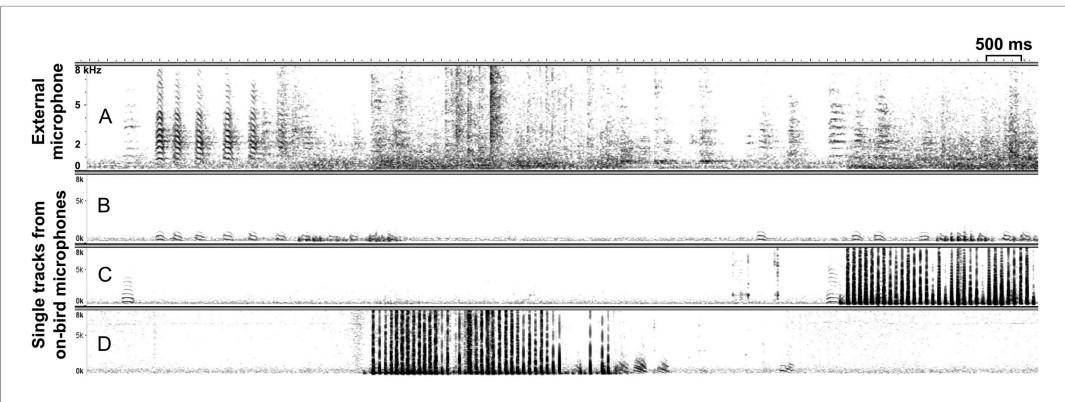

**Figure 7**. Synchronous external and on-bird recordings. Example spectrograms (x-axis: time [ms], y-axis: frequency [kHz]) of synchronous recordings with (**A**) group recording from external microphone without individual information and (**B**–**D**) individual recordings from three (out of eight) on-bird microphones. Dark vertical lines (in **C** and **D**) represent wing beats that hardly show up in the noisy external recording. Note the higher power in the low frequencies in the on-bird microphone recordings, compared to the external recording.

suggesting that an interaction pattern emerged that was pair specific and synchronous within each group. Two of the 10 pairs did not show this vocal interaction pattern (pairs L and F in *Figure 5A*) and also did not lay eggs.

In total, out of all possible combinations between all call types from all individuals (within-bird interactions excluded) only 6.5% resulted in significant interactions (see 'Materials and methods' for definition). 4.8% showed positive values, that is, the calls of one bird led to an increase in the calls of another bird, and 1.7% showed negative values, that is, the calls of one bird led to a decrease in the calls of another bird. Within pairs (n = 10), 9.2% of the possible interactions was significant, with 8.3% being positive and 0.9% negative. Further, the data suggest that vocal activity and the amount of vocal interactions decreased with progressing breeding stages, but the ratio of vocal interactions with the partner compared to those with other group members showed a fivefold to sixfold increase (from 0.74 when unpaired to 4.24 during incubation; *Figure 5—figure supplement 1*).

Within pairs, not all possible call-type combinations were used in significant positive vocal interactions; for example, distance calls were never used in combination with whines. The highest percentages of within-pair calling interactions took place in tets, stacks, and cackles (*Figure 5B*). Same-call interactions (bottom left to top right diagonal in *Figure 5B*) were not more common than interactions between different call types. However, same-call interactions were more symmetrical between females and males, and changed over the breeding stages almost in the same way for both sexes. In contrast, different call-type interactions were less symmetrical between the sexes. For example, tets were more likely to be answered by stacks when the responding bird was a female, and stacks were more likely to be answered by tets or cackles when the responding bird was a male, especially at the onset of nesting activities (*Figure 5B*). In this case, the asymmetries thus changed over the breeding stages, with a peak at 'nest inspection'. Breeding stage thus had an effect on different call-type combinations.

The number of positive within-pair calling interactions was not only related to reproductive state, but also to whether or not a pair succeeded in producing a clutch of eggs ('successful egg-laying'). The number of call-type combinations with significant interactions increased over reproductive stages for pairs that laid eggs, but failed to do so for pairs that did not lay eggs (*Figure 6*). This means that pairs involving in more call-type interactions at certain stages were more likely to produce a clutch of eggs. Successful pairs only shared significant interactions in few call-type combinations before nesting (1.4 ± 0.83 call-type combinations ±SD) and increased these interactions during the later nest stage (8.92 ± 4.41). Unsuccessful pairs, on the other hand, showed decreasing numbers and also higher levels of variation throughout the breeding stages (from 4.48 ± 3.35 to 1.25 ± 1.5, respectively), suggesting a less specific usage of call types in interactions.

## Discussion

Our study showed with individual-level resolution that call-based vocal communication of group-living zebra finches changed across reproductive stages. Using the temporal information encoded in call onsets, we found that the timing of calls was not random but instead occurred in significant vocal interactions between individuals of social groups. Both individual-level call-type usage and calling interactions between mates and other group members changed with breeding stages. The quality of calling interactions between pair members during several reproductive stages was correlated with successful egg-laying.

### Calling behaviour in social environments: new approach, new results

In songbirds, there is increasing evidence that not only song, but also calls can play a role in reproduction (*Groth, 1993*; *Marler, 2004*; *Elie et al., 2010*; *Ter Maat et al., 2014*). However, the usage and function of call repertoires, especially in group-living species, has been unknown, so far. The zebra finch is one of the prime model organisms for studies on vocal communication, especially with regard to song. Its vocal repertoire, including song and different call types, has been described in most detail by Zann's laboratory and field observations (*Zann, 1996*) to which we found various parallels in our data. For example, as suggested (*Zann, 1996*), loud distance calls occurred most before birds were paired or nesting, and cackles and whines increased at the onset of breeding activities (*Figure 4*, *Figure 4—figure supplement 1*). However, in most previous analyses of zebra finch vocal behaviour, technical limitations constrained a reliable separation between individual sound sources when birds behaved in social contexts involving direct contact with multiple individuals (*Zann, 1996*; *Elie et al., 2010*, *2011*). This was especially relevant when birds vocalised quietly and in close proximity to each other (*Elie et al., 2010*). Although it has been suggested that the vocal output of entire zebra finch groups depends on group structure (*Elie et al., 2011*) and that quiet calls may play a role in pair formation (*Zann, 1996*; *Elie et al., 2010*; *Ter Maat et al., 2014*), these previous studies did not investigate calling interactions inside groups with individual-level resolution. For instance, it had been stated that quiet tet calls are produced almost at all times, are not directed at specific individuals, and therefore do not stimulate specific replies (*Zann, 1996*). Instead, we found that tet calls did not occur randomly, but in meaningful interactions between individual birds.

Other studies recorded individual vocalisations, but resorted to strongly reduced social contexts and environmental enrichment, often coupled with frequent disturbances through bird handling (e.g., *Blaich et al., 1996*; *Vignal et al., 2004*; *Anisimov et al., 2014*). Therefore, despite slightly different vocalisation classification paradigms (*Vignal et al., 2004*; *Elie et al., 2011*; *Anisimov et al., 2014*; *Ter Maat et al., 2014*), between-study differences in the birds' vocal repertoire contents are most likely due to social context, as this can impact a multitude of physiological and behavioural aspects which in turn may be linked to vocalisations, for example, high rates of stack-call production in isolated birds (*Zann, 1996*).

Our setup allowed combining a species-relevant context with recording techniques (*Figure 1*, *Video 1*) that ensured longer-term individual recordings of freely behaving birds with infrequent bird handling. This enabled us to explore new aspects of vocal communication, including functional aspects of vocal interactions, as discussed below.

### Opportunistic breeding and vocalisations

Zebra finches are opportunistic breeders, and in arid habitats, they rely on short and unpredictable periods of rainfall to successfully rear their young. Therefore, birds need to start breeding immediately when environmental conditions permit (*Zann et al., 1995*; *Prior et al., 2013*), and may do so throughout the year, in the wild as well as in captivity (*Zann, 1996*; *Perfito et al., 2007*; *Perfito, 2010*). In our study, newly joined birds quickly formed pairs and increasingly engaged in reproductive activities which were correlated with increased concentrations of gonadal hormones (see Appendix 1). In parallel, shifts in reproductive stages were associated with changes in calling behaviour. First of all, at the individual level, the call repertoire changed, sometimes showing sex-specific patterns. Second, vocal networks were dynamic, and showed increasing differences between pair and group communication. The most synchronous and therefore most apparent changes in reproductive stages occurred at the sudden onset of nest material which was accompanied by nesting behaviours in most birds and was reflected by a pair-specific pattern lighting up in the vocal interaction matrices. Intriguingly, the quality of within-pair vocal interactions was associated with successful egg-laying. Although all pairs engaged in

nesting behaviours at some point, those sharing interactions in more call types during later nest stages were more likely to succeed in producing a clutch of eggs during the 3-week trials. In addition, the different levels of variation, especially before the onset of breeding activities (pre-nesting), suggest that successful pairs were more specific in their call-type usage in interactions, demonstrating the importance of call types in pair communication during the breeding cycle. Because in the wild, birds need to start breeding immediately with the unpredictable onset of rain (see above), such rapid changes in fine-tuned pair communication could be essential for successful reproduction. Our findings thus offer an additional aspect of opportunistic breeding behaviour by showing that changes in the environment, leading to changes in reproductive state, were accompanied by transient changes in the calling behaviour inside groups, involving a shift towards dyadic pair communication.

In zebra finches, song has been shown to be important for mate choice and pair formation but to lose significance once a stable pair bond is established (*Adkins-Regan and Tomaszycki, 2006*). Our findings not only support that calls and calling interactions between mates are important for pair formation (*Elie et al., 2010*; *Ter Maat et al., 2014*), but also suggest an important role for successful reproduction (egg-laying). In this species, song is produced only by males, and thus constitutes a unilateral signal. Calls, on the other hand, are produced by both sexes and can be exchanged bilaterally. Therefore, they have the potential to be used in mutual and more complex behavioural interactions supporting pair formation and synchronisation as well as pair-bond maintenance, as suggested for other mutual behavioural displays (reviewed by *Bradbury and Vehrencamp, 2011*). Hereby, some call types are used more frequently in vocal interactions than others. We therefore suggest that for rapid calling exchanges especially the soft and short tet, stack, and cackle calls may be more suitable than longer and more variable calls, such as whines.

It remains to be explored whether or how gonadal hormones affect calling behaviour across changing reproductive stages (see Appendix 1). Due to large differences in the temporal resolution of sampling methods for hormones and behaviour, a direct comparison between concentrations of circulating hormones and of call communication is difficult (see Appendix 1 for a more detailed discussion of hormones and calling dynamics in groups). Next to hormones, the decision of an initiating bird to produce a certain call and of a responding bird to answer with a certain call may depend on the specific context, on the behaviour of the other group members, or on the responder's previous experience (as suggested for social bonding in other bird species; *Vignal et al., 2004*; *Emery et al., 2007*). To date, we can only speculate on how such vocal interaction patterns are established in this songbird species. Learning might play a role in females and males, because both sexes initiated calling interactions and responded to calls in an increasingly synchronised pattern during progressing reproductive stages. Also, it remains to be investigated whether call-type usage between potential mates supports assortative mating (*Moravec et al., 2006*), or has a direct effect on fertility, as suggested for specific vocalisations in budgerigars (*Melopsittacus undulatus*) (*Brockway, 1965*, *1967*). Physiological and behavioural synchronisation of pair members have been suggested to play an important role in successful reproduction (*Wickler and Seibt, 1980*; *Cheng, 2003*; *Hirschenhauser et al., 2008*; *Ouyang et al., 2014*). To our knowledge, ours is the first study to demonstrate an association between intra-pair calling dynamics and successful egg-laying, thus highlighting the potential fitness relevance of calls and calling interactions in communication systems.

## Materials and methods

### Ethical note
Animal housing and welfare were in compliance with the European directives for the protection of animals used for scientific purposes (2010/63/EU). Protocols were approved by the Government of Upper Bavaria.

### Study design, housing and nest material
Between November 2011 and August 2012, four group trials of about 3 weeks each were carried out in succession on a total of 32 adult zebra finches (4 females and 4 males each) which were fully adult offspring from our breeding colony. Future group members of the opposite sex had never previously met, that is, had been raised in separate rooms, and siblings were not included in the same trials. We also ensured that birds had not bred before. Birds were individually recognised by one numbered and two coloured leg bands, and were kept in a 14:10 hr light:dark cycle with ad-libitum access to water and seeds and additional greens and egg-food.

The birds were caught from large same-sex aviaries and equipped with on-bird audio transmitter 'backpacks' (*Figure 1*). They were subsequently held in smaller same-sex aviaries (170 × 165 × 80 cm) for habituation to the backpacks before the beginning of the trials. Different habituation phases have been reported in similar studies, which is likely due to differences in the weight of the backpacks applied to the birds. Reported backpack weights on captive zebra finches ranged from 0.6 g (Adreani et al., 2015, in prep.) to 3 g (*Anisimov et al., 2014*), and resulted in habituation phases between a few days (1 day for movement, 4 days for call rates, Adreani et al., 2015, in prep.) up to about 2 weeks (*Anisimov et al., 2014*). Our backpacks weighed 1 g, and based on prior observation (unpublished data), we chose a 1-week habituation period to ensure birds had fully recovered from any behavioural effects of the backpacks and any associated handling.

Each trial began by placing the four males and the four females inside a large aviary (*Figure 1—figure supplement 1*; for timeline see *Figure 8*). This mixed-sex aviary (2 m × 2 m × 2.5 m) contained four large perches, four empty nest boxes, and eight antennas protruding vertically into the top area of the cage, thus offering additional perching opportunities (*Figure 1—figure supplement 1*). 1 week after the beginning of the recordings, nest material (coconut fibres and soft white lining) was provided and recurrently refilled. While the birds went through different breeding stages, they were blood-sampled for hormone analyses (see Appendix 1), and non-vocal behaviours were recorded during regular observations. Vocal behaviour was recorded almost continuously throughout the day by means of microphone telemetry, and exemplary morning sound recordings were analysed (see 'Materials and methods' on sound analyses). Behavioural observations and handling of technical equipment were carried out from behind a large, green curtain inside the experimental room, and to control for human disturbance, the observer quietly entered the room at least 10 min before each observation period. Animal care, nest checks as well as any handling of birds, took place outside of recording periods (see below).

For behaviour and hormone analyses, the data from all four trials (0, I, II and III) were used (n = 32). For sound analyses, only trials I, II, and III were used, because trial 0 served as a test run for the sound recordings and experimental design.

## Sampling

### Behavioural observations, nest checks and blood sampling
To establish which individuals formed pairs, to define their breeding stages, and to record agonistic and prosocial behaviour within the groups, behavioural observations were carried out for 30 min at least twice a day from day 1–4, at least once per day until day 10, and at least once every 2 days until

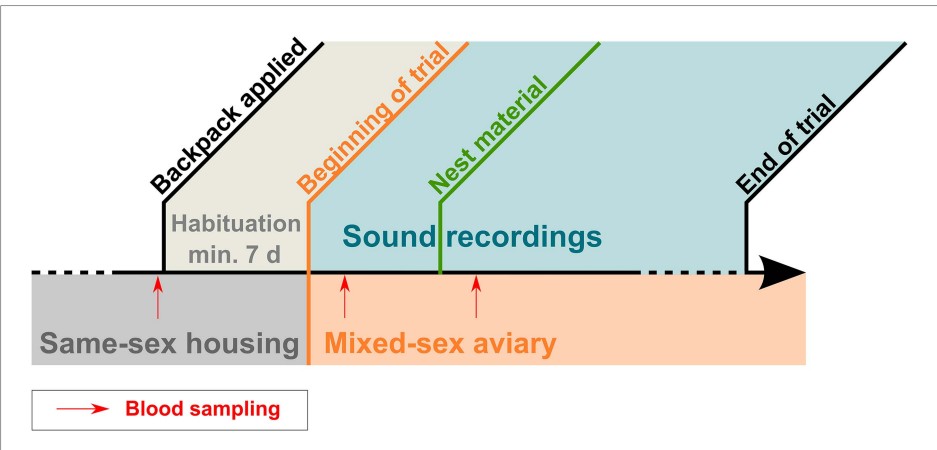

**Figure 8**. Timeline of trials. Timeline indicating housing conditions and approximate timing of backpack application, habituation phase (minimum 7 days), beginning of trial and sound recordings (ca. 3 weeks, see 'Materials and methods'), onset of nest material availability, end of trial. Of the continuous sound recordings, mornings (220 ± 20 min) of different days (differed between trials, not indicated in graph) equally representing birds' breeding stages were analysed. Blood sampling occurred three times (indicated by red arrows): before the beginning of the trials, 1 day after males and females were joined and 1 day after nest material became available.

birds had laid eggs (days 16–23). Prolonged close proximity and tactile contact, such as clumping and allopreening, between zebra finches has been shown to be indicators of pair formation (*Butterfield, 1970*; *Silcox and Evans, 1982*; *Tomaszycki and Adkins-Regan, 2005*). Therefore, the observer noted the behaviours listed in *Table 1* by coding them directly into a prepared spreadsheet on a laptop computer. Behaviours of short duration, for example, displacements, copulation solicitation, or entering the same nest box, were recorded as frequencies, while longer lasting behaviours, for example, clumping or incubating, were sampled once every 2 min and processed as durations. The given behaviour was recorded along with its time and location, and, if applicable, the identity of sender and receiver were noted (e.g., for displacements or allopreening).

Nest checks were carried out at least every second day between 7 and 9 hr from lights on when birds usually were relatively inactive. Nest checks included visual inspection of the nest material inside each nest box, counting the number of eggs and hatched chicks inside a nest, and individually marking them with a coloured pen. Sound recordings were paused during this time, and only those preceding nest checks were included in this manuscript. Behavioural observations were resumed a few hours after nest checks.

Blood samples to determine baseline hormone concentrations were taken at least 1 week before the beginning of the trial when the birds were still held in same-sex aviaries (see timeline in *Figure 8*). They were bled again 1 day after the beginning of each trial, that is, 1 day after being joined in the large mixed-sex aviary and 1 day after nest material had been added, a week later. To minimise any effects of blood sampling, we only included sound recordings from the same day as blood sampling, if they had been recorded before the procedure.

Birds were caught from the aviaries with hand-nets by two people and passed on to four other people who bled them. Hereby, the brachial vein was punctured using sterile syringes, and a small amount of blood (ca. 70 μl) was taken using a heparinised capillary tube, and subsequently transferred to a tube on ice. All blood samples were collected and stored on ice within 5–10 min of the initial disturbance of opening the door to the experimental room (245 ± 27 s). As soon as all birds had been processed, they were released back inside the aviary. Within 15 min, all tubes were brought to the lab where they were centrifuged at 3000 rpm at room temperature for 10 min. Plasma was separated from the blood cells and stored at −80°C.

## Vocalisation recordings

To capture individual vocalisations in a group setting, we used microphones mounted directly on the birds (on-bird microphones), and hereby chose microphone transmitters (Sparrow System, Fisher, III, USA). This provided a number of practical advantages in this setting, compared to using loggers (*Anisimov et al., 2014*), some of which we explain in the following. Due to the unified external recording process when using transmitters (see below), it was possible to gain recordings from all individuals simultaneously and in temporal synchrony (*Figure 7*) within each group. Thus, birds could be housed in a relatively large three-dimensional environment (2 × 2 × 2.5 metres) including nest boxes and perches at different heights (*Figure 1—figure supplement 1*, *Video 1*), because transmitters do not require synchronisation to an external, for example, visual signal (*Anisimov et al., 2014*). In addition, our devices—containing a miniature microphone, a radio transmitter and a button cell battery—were light (1 g), and batteries were replaced only after 6–10 days, which strongly reduced the amount of disturbances associated with bird handling, compared to previous studies (e.g., *Vignal et al., 2004*; *Anisimov et al., 2014*).

Devices were fixed on the back of the birds by a thin silicon harness, with straps around the neck and abdomen (*Figure 1*). In a different study (*Anisimov et al., 2014*), additional accelerometers were used to distinguish between vocalising individuals. However, as previously described (*Ter Maat et al., 2014*), directing the miniature microphone towards the bird's body, that is, facing inwards (*Figure 1*) instead of facing outwards, strongly reduces the probability of recording vocalisations of non-focal individuals, because microphones are attached as closely as possible to the sound source, and external sounds are dampened by the backpack. In addition, due to differences in spectral properties (*Figure 7*; also see 'Materials and methods' section on sound analyses), any (rarely) recorded non-focal birds' calls were easily detected and excluded during the sound analyses.

Backpacks were modified for females in such a way that the originally dorso-caudally protruding antenna was incorporated into the silicon loop to ensure that copulation remained possible. As this procedure tended to decrease radio signal strength, it was not done for males who exhibited normal

copulation behaviour despite a protruding antenna. Eight communications receivers (AR8600, AOR, USA) set up outside the aviary were connected to the eight antennas inside the aviary and received the respective AM-modulated radio signals (transmission frequency: 375–380 mHz) coming from the microphone transmitters on the backs of the eight birds. A 16-channel AD-converter (Sonic Core A16 Ultra) was used to digitise the analogous signals. Using a custom multi-channel programme (ASIO Rec, Markus Abels, MPIO Seewiesen), we recorded all eight channels at a rate of 44,100 Hz and stored them as uncompressed files (.wav) on a single computer to ensure temporal synchrony between them. Recordings were automatically started and stopped for all eight birds simultaneously and were made for at least 4 hr in the mornings and evenings every day. An exemplary subset of the recordings was analysed for this manuscript (see 'Materials and methods' on sound analyses).

## Analyses

### Reproductive stages

To objectively describe pairs' breeding stages, two sets of categories were created from the 30-min observations and nest checks (*Table 2*, *Figure 2*). The first one ('breeding stages') was in greater detail and was summarised to form the second one ('nest stages') which was required for statistical analyses.

For the evaluation of hormones at different reproductive stages, we used the results from 95 blood samples collected at the three time points (baseline n = 31, 1 day after joining of birds n = 32, 1 day after adding of nest material n = 32) from the 16 male and 16 female zebra finches. Using a modified version (*Goymann et al., 2008*) of the radioimmunoassay method established by *Wingfield and Farner (1975)*, we determined concentrations of testosterone (T), dihydrotestosterone (DHT), oestradiol (E2), and progesterone (P4). Samples were run in duplicates for each individual and measured in a single assay. The intra-assay variation of extracted chicken pools was 12.4% for T, 17.8% for DHT, 29.7% for E2 and 15.5% for P4, respectively. Steroid extraction efficiency (mean percentage ± standard deviation [SD]) was 70.9 ± 2.7 (T), 73.7 ± 4.3 (DHT), 59.9 ± 4.0 (E2), 55.9 ± 11.0 (P4), and the detection limits measured in pg per tube were 0.39, 0.43, 0.2, and 2.38, respectively. As 77 out of the 95 E2 samples were below the detection limit, this hormone was excluded from further statistical analyses.

### Sound analyses

Due to technical issues, we had to reject the recordings from two out of four males from trial I. Six (trial I) or seven (trials II and III) sound files per trial were chosen post hoc according to the following rationale for analysis: The respective recordings were started within 2 hr of lights on, and continued for a maximum of 4 hr (i.e., finished before birds' midday). To minimise any external effects on the vocalisations, sound files qualifying for analysis also preceded any major disturbances (e.g., nest checks or blood sampling, see below and Appendix 1). Because breeding stages were not synchronous between birds and groups, sound files were chosen for analysis so as to give a relatively balanced representation of all pairs' breeding stages. In total, we analysed 146 sound files (67 for males, 79 for females) coming from up to 10 male and 12 female zebra finches (10 pairs) in the three different trials, summing up to about 535 hr of sound recordings.

In all of the following steps, temporal information was retained between individual recordings. To remove low frequency noise, for example, originating from external technical equipment, we used a high-pass filter (200 Hz cut-off frequency; apple Audio Unit) in Amadeus Pro (2.0.5, HairerSoft, UK) on all recordings. We used custom programs (available at https://github.com/ornith) to detect, extract and time-stamp sound events, to identify and remove noise (e.g., wing-flapping or noise from degraded radio signal) and non-focals' vocalisations, to automatically define onset and offset of syllables, and to classify them via a k-means clustering paradigm (*Ter Maat et al., 2014*). During this procedure, the (rare) vocalisations that had been recorded from non-focal individuals were easily detected and removed, because they show less power in the low frequencies, compared to the focal birds' recordings (*Figure 7*). If a syllable's start or end had been defined inaccurately (e.g., due to overlapping noise) and its type was therefore not identifiable or its actual onset did not coincide with the onset of the noisy interval, it was manually deleted. Thus, we retained as many syllables as possible for the analysis but reduced potential confounding artefacts in the temporal information of syllable onsets. As zebra finch males may incorporate some of their calls into their song (*Zann, 1996*), we arranged all syllables in their order of occurrence before clustering and were thus able to separate similar vocalisation types used in song bouts from those occurring as single calls. The program compares each sonogram in an automatic process with respect to the following nine temporal–

spectral parameters: duration, mean frequency and standard deviation (SD), mode frequency and SD, Wiener entropy and SD, first peak and SD. The output was manually refined by visual inspection of the spectrograms so as to reach a classification of different call types according to the descriptions in *Zann (1996)*. As is described there, cackles, arcs, and whines are associated with the onset of nesting activities and might blend into each other—with cackles being the shortest and the most inflected ones and whines the longest and the least frequency-modulated versions. Arcs lie in between these two types in terms of occurrence, duration, and frequency modulation. Therefore, we decided to use only the calls at the ends of the continuum, namely 'cackles' and 'whines' and to discard the 'arcs' from further analyses. If it was not possible to unambiguously assign single calls to a category, they were moved to a different class which was not included in further analyses. As the aim of our study was to investigate the role of calls in zebra finch groups and the clustering paradigm was optimized for calls, we did not include song in our statistical analyses. Instead, we chose 'distance calls', 'stacks', 'tets', 'cackles', and 'whines' (*Figure 3*) because they were the most frequently used and easily classified call types. We found the same call types for males and females despite potential sex-specific differences in spectral features, as for example in the loud distance calls that contain learned frequency modulations in males but not in females (*Simpson and Vicario, 1990*) (*Figure 3A*).

The number of vocalisations was calculated for each vocalisation type for each bird for each recording. As the durations of recordings were not always identical (220 ± 20 min), these values were extrapolated to the longest recording (243 min), to avoid working with fractions. For the overall calling activity, we summed up these extrapolated numbers of vocalisations in the five call types for each bird and recording period. In addition to this absolute measure of call usage, we divided the number of calls per type by the total sum of calls to obtain the relative proportion of a bird's call types for each recording period.

## Vocal interaction analyses

Custom software written in C++ [available at https://github.com/ornith] was used to merge up to eight single result files per recorded interval. The data were thus stored in a large text file containing the starting point of each vocalisation within the recorded period, its syllable type and the sender's identity. In R, (3.1.1, R Development Core Team) these tables were concatenated and supplemented with information on the respective breeding stages and hormone data. Previous examinations (unpublished data) of spontaneous calling interactions in zebra finch groups revealed that a large proportion of vocalisations occurring within a few seconds of another bird's calls in fact occurred within half a second. As described in *Ter Maat et al. (2014)*, for each recording period, vocalisation starting points of two birds were aligned to each other in peri-stimulus time histograms (PSTHs) to gain cumulative sums from which a vocal association index can be calculated. We, too, chose a time window of interest of 4 s before and 4 s after call onset for our PSTHs with a binwidth of 50 ms (i.e., 160 bins in total). The number of calls in the first 0.5-s bins ($N_{base}$) served as baseline and those in the 0.5-s bins after call onset ($N_{response}$) as the response. The correlation index of a 'response' for each set of two birds relative to each other was calculated as follows:

$$R_{response} = \left( N_{response} - N_{base} \right) / \left( N_{response} + N_{base} \right).$$

Using a 95% confidence interval, we determined whether the occurrences of calls in a specific combination of individuals and call types could be considered random or as a 'significant interaction'. This will be referred to as a non-zero correlation index (see below for statistical analyses). In this way, vocal activity was considered in calculating the strength of significant interactions. For example, a strong positive correlation in a dyad (turquoise square in *Figure 5A*) is not a result of high vocal activity, but of accurate timing of vocalisations between two individuals. Likewise, a value of 0 (grey square in *Figure 5A*) means that in this particular dyad, the significance criterion was not met (there was no significant vocal interaction), but does not mean that no vocalisations took place. 50 ms after call onset were removed from the analyses to avoid crosstalk (i.e., in case the same birds had been recorded by both backpacks). Data were considered valid only if each bin contained on average one or more events (i.e., total >160 events).

From the correlation indices, we generated confusion matrices for all combinations of the five call types for all combinations of birds in a group (40 × 40 possible combinations for trials II and III and 30 × 30 matrices for trial I due to missing values from two males) (*Figure 5A*). We did not find

significant interactions for all groups on all days: on 2 out of the 20 analysed days, only within-bird interactions occurred frequently enough to exceed the confidence interval.

To further analyse these values for the different breeding stages and call type combinations, we only used within-pair interactions (trials I–III), because group members did not always have the same breeding stages on the same days. For within-pair interaction data, we calculated the percentage of positive to all possible call combinations for males and females of each pair in each breeding stage and for each call-type combination. We also calculated the percentage of positive interactions for pairs that laid eggs during the trials and those that did not (overall, when unpaired, during nest inspection and nest building). To investigate pair specificity of vocal interactions, we calculated the percentage of possible interactions at different breeding stages with the partner and the other group members (n = 16 birds, from trials II and III, due to incomplete data in trial I).

## Statistical analyses

Statistical analyses were performed using R version 3.1.1 (R Development Core Team). To account for our study's repeated measure design, we used mixed models to analyse hormones (see Appendix 1) and vocal activities over the breeding cycle (dataset Gill et al. available at http://datadryad.org/10.5061/dryad.vt69s). An information theoretic approach was used to analyse the relationship between pair call-type combinations and successful egg-laying (see below). As frequentist analyses do not allow an exact calculation of the degrees of freedom (*Bolker et al., 2009*), we chose a Bayesian statistical approach (with uninformed priors). This allows drawing inferences as meaningful differences between groups by evaluating the ranges of their 95% credible intervals (CrI, range of group means, at 0.95 certainty). Thus, we ran linear mixed-effects models (lmer) using the R package 'arm' (version 1.7–05) on the different dependent variables, with bird identity as a random factor.

Posterior means and CrI were calculated using the function sim (running 5000 simulations) and were compared to find meaningful differences between groups. Therefore, if the mean estimate of one group did not overlap with the CrI of another group, it could be inferred that these groups differed from each other. Posterior means and CrI were plotted alongside boxplots of the raw data for vocal activities and hormones in relation to nest stages, and differences are indicated by different letters above each box (*Figure 2—figure supplement 1*, *Figure 4*). For reasons of clarity, we provide a table containing these values for the more detailed breeding stages (*Figure 4—figure supplement 1*, Appendix 2, *Appendix 2—table 1*).

Graphical methods (plotting model residuals) were used to evaluate model fit and whether model assumptions were met. To improve residual distributions, some of the numeric-dependent variables were natural-log or square-root-transformed (indicated in graphs). To describe the variance explained by our models, we provide F values (ratio of between-group to within-group variance) and marginal and conditional R-square values that range from 0 to 1 and describe the proportion of variance explained by the fixed and by the fixed and random effects combined, respectively. They were calculated according to *Nakagawa and Schielzeth (2013)*. As we expected to find marked differences both in the hormone concentrations and in the vocalisations of males and females, sexes were analysed separately to keep the models as simple and powerful as possible. Including trial ID as a further explanatory variable did not affect the outcome of any statistical test, and therefore, it was removed from all models.

## Hormones

To find out about the relationship between hormone levels and nest stages, we ran a model for each of the three hormones (log-transformed) with a four-factor variable including baseline levels and the three nest stages (pre-nesting, early, and later nest stages) as fixed effect and bird identity as random factor.

## Vocalisation types and breeding activity

To get an overview over the usage of different call types during different phases of the breeding stage, we ran a model for each vocalisation type on the frequency of vocalisations per recording period (extrapolated to the longest recording duration) in relation to the three-factor variable nest stage (pre-nesting, early, and later nest stages), again with bird identity as random factor. For a more detailed impression of the relative changes in call usage according to breeding stages, we ran another set of models on the proportion of vocalisations of one call type in relation to all call types over the more detailed variable 'breeding stage' as fixed effect (Appendix 2). This was done on all vocalisation types separately and always included bird identity as a random factor.

### Within-pair interactions, breeding activity and breeding success

As data points were few (10 pairs, subdivided into groups), we decided on a compromise between pure description of the data and complex models accounting for repeated measures to get an idea of whether calling interactions between pair members at different stages of the breeding cycle were related with whether or not pairs would lay eggs within the 3 weeks of the trial (pair success). To be able to compare successful and unsuccessful pairs (*Figure 6*), we only included breeding stages before the egg-laying phase, that is, until 'nest building', as 'egg-laying' and 'incubation', by definition, only occurred in 'successful' pairs. If the same pairs were sampled multiple times per breeding stage, mean values were calculated. We then performed model selection according to *Anderson (2008)* on the number of positive interactions in relation to nest stage and success, assuming a poisson distribution. After checking model assumptions, we calculated corrected AIC values (cAIC) to select the best fitting model (lowest cAIC). This turned out to be the model including an interaction between the two explanatory variables.

## Acknowledgements

We would like to thank Hannes Sagunsky and Markus Abels, Susanne Seltmann, Ingrid Schwabl and especially Lisa Trost for technical support, and Monika Trappschuh for helping with hormone analyses. Further, we thank Nicole Geberzahn and Wolfgang Wickler for their helpful comments on previous versions of the manuscript. We thank the International Max Planck Research School (IMPRS) for Organismal Biology for training and support.

## Additional information

### Funding

| Funder | Author |
| --- | --- |
| Max-Planck-Gesellschaft (Max Planck Society) | Lisa F Gill, Wolfgang Goymann, Andries Ter Maat, Manfred Gahr |

The funder had no role in study design, data collection and interpretation, or the decision to submit the work for publication.

### Author contributions

LFG, Conception and design, Acquisition of data, Analysis and interpretation of data, Drafting or revising the article, Contributed unpublished essential data or reagents; WG, Conception and design, Acquisition of data, Analysis and interpretation of data, Drafting or revising the article; ATM, MG, Conception and design, Analysis and interpretation of data, Drafting or revising the article, Contributed unpublished essential data or reagents

### Ethics

Animal experimentation: Animal housing and welfare were in compliance with the European directives for the protection of animals used for scientific purposes (2010/63/EU). Protocols were approved by the Government of Upper Bavaria under the licenses 55.2-1-54-2531-25-09 and 55.2-1-54-2532.8-76-11.

## Additional files

### Major dataset

The following dataset was generated:

| Author(s) | Year | Dataset title | Dataset ID and/or URL | Database, license, and accessibility information |
| --- | --- | --- | --- | --- |
| Gill LF, Goymann W, Ter Maat A, Gahr M | 2015 | Data from: Patterns of call communication between group-housed Zebra finches change at different stages of the breeding cycle | http://datadryad.org/ 10.5061/dryad.vt69s | Available at Dryad Digital Repository under a CC0 Public Domain Dedication. |

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

## Appendix 1

# Hormones in relation to reproductive stages.

The steroid hormones testosterone, dihydrotestosterone (DHT), oestradiol, and progesterone are known to be associated with different reproductive stages (*Wingfield and Farner, 1993*). Therefore, we analysed plasma steroid concentrations of our birds at different time points to supplement behaviourally classified reproductive stages with information on the physiological status (for 'Materials and methods', see main manuscript).

Reproductive status was correlated with hormone concentrations. Testosterone increased for females and males over the course of each trial: in females, testosterone concentrations were higher for the later nest stage compared to baseline levels ($F_{stage} = 1.6$, $R^2_{marginal} = 0.06$, $R^2_{conditional} = 0.43$). Males expressed higher testosterone concentrations in early and later nest stages compared to the baseline. Further, at the later nest stage, males had higher testosterone levels compared to the pre-nesting stage ($F_{stage} = 8.36$, $R^2_{marginal} = 0.21$, $R^2_{conditional} = 0.65$; *Figure 2—figure supplement 1*). Dihydrotestosterone (DHT) did not change for females during the course of each trial ($F_{stage} = 0.28$, $R^2_{marginal} = 0.01$, $R^2_{conditional} = 0.37$; *Figure 2—figure supplement 1*). For males, there were no differences in DHT concentrations between baseline and pre-nesting, or between early and late nest stages. DHT concentrations were, however, higher at the early and late nest stages than at baseline and pre-nesting ($F_{stage} = 5.46$, $R^2_{marginal} = 0.18$, $R^2_{conditional} = 0.57$; *Figure 2—figure supplement 1*). Progesterone changed for both, females and males. In females, progesterone levels were higher during the later nest stage compared to all other stages ($F_{stage} = 9.39$, $R^2_{marginal} = 0.30$, $R^2_{conditional} = 0.52$; *Figure 2—figure supplement 1*). In males, progesterone concentrations were higher during early nest stage compared to the baseline ($F_{stage} = 2.26$, $R^2_{marginal} = 0.13$, $R^2_{conditional} = 0.16$; *Figure 2—figure supplement 1*).

Newly joined birds quickly formed pairs and increasingly engaged in reproductive activities. These were correlated with increased concentrations of plasma testosterone in females and of androgens (DHT and testosterone) in males. Our results are thus in line with studies showing that changes in the social and abiotic environment—for example the presence of receptive females or of nest material—are reflected by changes in plasma testosterone levels and the emergence of mating behaviours (*Riters and Alger, 2011*). In addition, progesterone, associated with pair-bonding (*Smiley et al., 2012*) and reproductive behaviours in females, has been shown to be elevated especially around the egg-laying phase (*Wingfield and Farner, 1993*; *Sockman and Schwabl, 1999*), further corroborating our results. Whether the differences in calling behaviour over the changing reproductive stages reflect direct effects of gonadal hormones remains to be explored. Androgens are known to affect the abundance of vocalisations of many vertebrate taxa (review in *Bass and Remage-Healey, 2008*), and in our study, changes in male androgen levels paralleled changes in particular call types across reproductive stages (decreasing distance calls and increasing cackles and whines). In females, there were no similar patterns in gonadal hormones and call usage, as only progesterone increased during the later breeding stage. Therefore, if gonadal hormones affected call usage in females and males, sex-specific hormonal profiles would mediate such dynamics. It is possible that gonadal hormones affected auditory preferences or auditory–motor interfaces, leading to a change in call usage via changed neural responsiveness (*Metzdorf et al., 1999*; *Avey et al., 2008*; *Remage-Healey et al., 2010*). However, it is unlikely that the slowly fluctuating gonadal hormones determine the closely timed vocal interactions. Rather, they may accompany changes in reproductive states at a larger temporal scale. Unfortunately, a direct comparison between concentrations of circulating hormones and of call communication cannot yet be drawn, due to extreme differences in the temporal resolution of reliable sampling methods available to date for hormonal data and for behavioural and acoustic

information. Nevertheless, our findings show that hormones were correlated with changes in reproductive stages, which, in turn, were correlated with calling behaviour on various levels.

## Appendix 2

### Call types and more detailed breeding stages.

To find out in more detail how the relative usage of vocalisation types might change during which part of the breeding cycle, we examined the effects of sex and of the seven detailed breeding stages on the proportion of vocalisations for each vocalisation type, that is, the number of vocalisations of a specific vocalisation type during a recording divided by the number of all vocalisation types during the same recording period. Here, we found an interaction between sex and breeding stage for all vocalisation types except for distance calls (**Figure 4—figure supplement 1**). In males and females, distance calls were used less in all other breeding stages than when birds were 'unpaired'. In females, they also occurred more frequently when birds were during the 'no nest' stage (paired but without own nest) than during 'nest building' and 'female egg-laying' ($F_{stage} = 8.60$, $F_{sex} = 0.29$, $R^2_{marginal} = 0.32$, $R^2_{conditional} = 0.40$). Tets showed two peaks during the breeding stages for females: one when birds were paired without own nest ('no nest') and one during 'egg-laying' ($F_{stage} = 2.69$, $R^2_{marginal} = 0.08$, $R^2_{conditional} = 0.20$). In males, tets occurred at intermediate levels during the 'unpaired' and 'no nest' stages, reached a peak during 'nest inspection' and then decreased again ($F_{stage} = 5.36$, $R^2_{marginal} = 0.22$, $R^2_{conditional} = 0.39$). Female stacks occurred frequently during 'unpaired', 'territorial' and 'nest inspection' and were reduced during 'no nest' and 'nest building' which was followed by another increase ($F_{stage} = 2.27$, $R^2_{marginal} = 0.09$, $R^2_{conditional} = 0.16$). For males, the proportion of stack calls was lower than for females and showed less prominent differences between breeding stages: there was a decrease over the course of the breeding stages. The highest peak, however, was at 'female egg-laying' ($F_{stage} = 2.09$, $R^2_{marginal} = 0.08$, $R^2_{conditional} = 0.34$). Females used cackles least when they were 'unpaired' and most during 'nest building' (mean estimate lay on the higher credible interval of 'egg-laying') ($F_{stage} = 4.12$, $R^2_{marginal} = 0.30$, $R^2_{conditional} = 0.41$). In males, the number of cackles gradually increased over the breeding stages until 'territorial' and 'nest building' was reached, and then gradually decreased back to 'unpaired' values ($F_{stage} = 4.67$, $R^2_{marginal} = 0.11$, $R^2_{conditional} = 0.52$). A similar pattern was found for whines in females ($F_{stage} = 1.96$, $R^2_{marginal} = 0.35$, $R^2_{conditional} = 0.11$) and in males ($F_{stage} = 3.16$, $R^2_{marginal} = 0.05$, $R^2_{conditional} = 0.63$): there was an increase in the proportion of whines over the breeding stages, reaching maximum values at 'territorial' and 'nest building', followed by a decrease back to 'unpaired' levels (**Figure 4—figure supplement 1**).

**Appendix 2—table 1**. Bayesian estimates and credible intervals (CrI) for the proportion of call types in females and males in relation to the detailed Breeding stages

| Call type | Breeding stage | Females | | | Males | | |
|---|---|---|---|---|---|---|---|
| | | Estimate | Lower CrI | Upper CrI | Estimate | Lower CrI | Upper CrI |
| Distance | Unpaired | 0.2570 | 0.1895 | 0.3218 | 0.2472 | 0.1818 | 0.3129 |
| | No nest | 0.1231 | 0.0629 | 0.1841 | 0.1133 | 0.0519 | 0.1765 |
| | Nest inspection | 0.0620 | −0.0151 | 0.1380 | 0.0522 | −0.0286 | 0.1293 |
| | Nest defence | 0.0565 | −0.0047 | 0.1202 | 0.0467 | −0.0150 | 0.1111 |
| | Nest building | 0.0351 | −0.0194 | 0.0895 | 0.0252 | −0.0308 | 0.0824 |
| | Egg-laying | 0.0335 | −0.0354 | 0.1048 | 0.0236 | −0.0502 | 0.1001 |
| | Incubation | 0.0920 | 0.0109 | 0.1712 | 0.0822 | −0.0014 | 0.1630 |
| Tet | Unpaired | 0.2038 | 0.1398 | 0.2680 | 0.2038 | 0.1398 | 0.2680 |
| | No nest | 0.1820 | 0.1233 | 0.2409 | 0.1820 | 0.1233 | 0.2409 |
| | Nest inspection | 0.3073 | 0.2278 | 0.3888 | 0.3073 | 0.2278 | 0.3888 |
| | Nest defence | 0.1073 | 0.0464 | 0.1659 | 0.1073 | 0.0464 | 0.1659 |
| | Nest building | 0.1370 | 0.0826 | 0.1902 | 0.1370 | 0.0826 | 0.1902 |
| | Egg-laying | 0.0977 | 0.0189 | 0.1790 | 0.0977 | 0.0189 | 0.1790 |

*Appendix 2—table 1. Continued on next page*

*Appendix 2—table 1. Continued*

|  |  | Females |  |  | Males |  |  |
| --- | --- | --- | --- | --- | --- | --- | --- |
| Stack | Incubation | 0.0902 | 0.0023 | 0.1803 | 0.0902 | 0.0023 | 0.1803 |
|  | Unpaired | 0.4061 | 0.2661 | 0.5393 | 0.1248 | 0.0636 | 0.1823 |
|  | No nest | 0.2299 | 0.1034 | 0.3530 | 0.1040 | 0.0487 | 0.1620 |
|  | Nest inspection | 0.3754 | 0.2212 | 0.5271 | 0.0695 | −0.0029 | 0.1444 |
|  | Nest defence | 0.3840 | 0.2571 | 0.5097 | 0.0941 | 0.0381 | 0.1514 |
|  | Nest building | 0.2133 | 0.1081 | 0.3151 | 0.0625 | 0.0109 | 0.1127 |
|  | Egg-laying | 0.2580 | 0.1220 | 0.3924 | 0.1646 | 0.0967 | 0.2360 |
|  | Incubation | 0.3493 | 0.1784 | 0.5183 | 0.0976 | 0.0166 | 0.1795 |
| Cackle | Unpaired | 0.0409 | −0.0658 | 0.1463 | 0.1101 | −0.0076 | 0.2278 |
|  | No nest | 0.2215 | 0.1288 | 0.3164 | 0.2114 | 0.0961 | 0.3258 |
|  | Nest inspection | 0.2160 | 0.1025 | 0.3335 | 0.2278 | 0.0865 | 0.3613 |
|  | Nest defence | 0.2771 | 0.1783 | 0.3729 | 0.2850 | 0.1721 | 0.3918 |
|  | Nest building | 0.2914 | 0.2118 | 0.3767 | 0.3477 | 0.2422 | 0.4507 |
|  | Egg-laying | 0.2002 | 0.0963 | 0.3102 | 0.2007 | 0.0631 | 0.3399 |
|  | Incubation | 0.1877 | 0.0672 | 0.3126 | 0.0839 | −0.0674 | 0.2399 |

