## [Decision Letter]

Thank you for submitting your work entitled “Patterns of call communication between Zebra finches at different stages of the breeding cycle” for peer review at *eLife*. Your submission has been favorably evaluated by Ian Baldwin (Senior Editor) and three reviewers.

The reviewers have discussed the reviews with one another and the editor has drafted this decision to help you prepare a revised submission.

All reviewers concurred that the use of telemetric acoustic recordings in a naturalistic social environment of a breeding group was novel and if you could do a better job of presenting what was interesting about your discoveries, this could be an *eLife* paper. The three reviewers provided a number of very helpful suggestions/concerns to improve the impact, presentation and defensibility of your claims. These were discussed and after some modifications, are listed below.

Reviewer 1:

Individual-level changes:

The paper addresses individual-level changes in vocal output using telemetric microphones to isolate and track vocalizations from individually identified birds. These conclusions, however, are based on the undefended supposition that the microphones mounted on each bird record only vocalizations generated by that bird alone. However, Anisimov et al. (2014; doi:10.1038/nmeth.3114), utilized telemetric microphone recordings in addition to accelerometer recordings mounted on individual birds and find that the accelerometer, not the microphone, reliably isolated vocalizations from individual birds. Steps need to be taken to defend the premise that the individual-level changes are truly generated by one bird and do not reflect simple changes in proximity to other birds, which are likely to change throughout the course of the experiment. This is an important point as this is what gives the paper its novelty. Providing a histogram of average syllable power recorded from each microphone would allow a clearer rejection of the non-self-generated signal. These histograms would ideally reflect a clear bimodal distribution and would allow the authors to more clearly assert their claims.

Control for experimental disturbances:

The authors present novel insights into vocal communication in the more ethologically relevant context of social housing. It would be useful to control for the effect of backpacks or other disturbances (such as nest checks and blood draws) on vocalizations. A useful control for the former would be to record vocalizations with a house microphone from birds with and without backpacks. [3] show a transient 2-week suppression in vocalizations upon fitting the backpacks.

Reviewer 2:

The Methods would benefit greatly from a visual timeline for readers to fully and quickly appreciate the timing and sequence of cage transfers, blood samples, recordings, etc. At present, the methods for the housing section of the study design are confusing. For example, one interpretation in the subsection “Study design, housing and nest material” is that pairs were separated after they formed and that the animals were held in single-sex aviaries to get used to the telemetric backpacks. This is unlikely, since the separation of pair mates is a significant stressor. Alternatively, if the authors intended information on the study set-up to appear prior to the information on study group background, that would be in keeping with a logical, chronological order of the study. Similarly, it is not until later in the section that we learn that the acoustic recordings were not continuous and were restricted to periods of the day. More detail about the recording time intervals is essential, since circadian factors can influence vocalizations in this and other songbird species.

It is not clear whether the external microphone was used as a reference to gauge the quality of the on-bird microphone recordings. This is important because the harmonic structure of vocalizations is much more apparent up to 8 kHz (Figures 3 and 8) in the sonograms from the external microphone, whereas by contrast the sonograms from the on-bird microphones show substantial frequency losses above 2 kHz. This indicates that there is a significant frequency filtering by the on-bird microphones and/or potential filtering by the placement of the microphone on the back of the bird (i.e. muffling by feathers/body). This is especially stark when comparing these recordings to Zann's original 1996 work on zebra finch calls. Zann's work clearly shows that calls have significant acoustical energy up to 8-9 kHz that may carry information about call identity/category. This therefore bears significantly on the experimenter/automated assignment of calls recorded by the on-bird microphones to different call categories.

The A/D board digitization rate (44 kHz) is good for the quality of the sound recorded, but in this case equally important is the transmission frequency of the on-bird radios. This needs to be provided, although it is likely to be in the megahertz range (the manufacturer's website, sparrow system, is currently unavailable). For example, if the on-bird radios are transmitting at low frequencies (i.e. under 22 kHz), this would reduce the quality of the recordings substantially, and may explain some of the frequency filtering noted above.

The extraction efficiency for the plasma steroid extraction should be provided.

One of the expectations I had going into the manuscript was that the authors had the opportunity to characterize the behavioral ‘predictors’ of individual call types. That is, they are conducting behavioral observations while also monitoring the vocalizations of individuals. If calls are acting as signals of behavioral intent or responsiveness, the results in this study could provide a much deeper look at vocal and social interactions than what is thus far reported in the study and what is known in this species.

Reviewer 3:

The study has challenges that I think need to be fixed. The biggest are that: 1) it under-sells the findings on call interactions in the abstract and beginning of the paper, 2) it over-sells being the first to use such microphone telemetry to record individual vocal interactions in songbirds, and 3) although some of the statistical analyses is based on the latest tools, it not presented clearly enough and some of the findings don't seem to be different as the text indicates. I highlight examples of these three points below.

1) The result mentioned in the Abstract is too non-specific, to the point that it does not say much. I think that several of these sentences should be substituted with ones that are more specific about what happened to the calls in individual pairs of animals, and then a more general conclusion statement about the overall conclusion of the paper – could be that use of this technology allowed them to identify previously unappreciated complex vocal communication interactions between a population of freely behaving birds. I think the authors should begin the Results section of the paper with some more of the exciting findings. Put the hormone findings last, and keep the order of everything else the same.

2) The authors write several long sentences about how their approach is the first, which almost any long sentence with a lot of qualifiers would be the first of its kind. However, one of the papers they cited (Anisomov et al., 2014 in Nature Methods) is the first real study that used such a telemetry approach to record vocal communication interactions between freely behavior songbirds. The current Gill et al. paper is the second that I am aware of. It looks like both studies were started independently. Nevertheless, the authors should acknowledge the similar approach used in Anisomov and compare and contrast with their approach, at least in the Methods section of the Gill et al. paper. The greater advance that the Gill et al. makes is that they performed and learned a lot more about the biology and behavior of the animals than did Anisomov et al. For that reason too, I think this paper is making an advance.

3) The box plots in Figures 1 and 3 and Figure 4—figure supplement 1 have a lot of information, which is not explained well enough. Most readers will look to see if there is overlap in the 95% CI boxes, and then assume that there are nearly no differences among groups. The authors should point the reader to the means being different (thin horizontal black line) in the legend. It is not clear to me what the vertical Bayesian CI gain is for this analysis? Why did the authors perform this statistic and add it to the box plots? Would the comparisons be significant without doing the Bayesian analyses and using standard repeated measures ANOVA? The letters a, b, and c in the box plots for statistical differences need a better explanation of what group is each letter comparing. The authors should also explain on first use in the results what does F, and R^2^_marginal_ vs. R^2^_conditional_ mean. This is not a typical statistic used in papers, and they cite it as something new in 2013 in the Methods, but without any explanation.

Specific important comments:

The authors use the word “natural” throughout the paper, and in the Introduction imply that they will study natural calling behavior, which they say little has been done before on. I am not aware of unnatural experiments. In addition, the authors perform experiments indoor, not in the natural wild environment, like most other investigators have. So I would not exactly consider this a natural experiment. This is another over-sold point, so much so that they attack some of the prior literature unnecessarily.

I like Figure 5, which is a novel view to me about displaying behavioral interactions. But it can use more explanation in the Methods or in the legend. Does the position of the data dots (turquoise to red) within the small tiny grey squares mean anything? The authors should emphasize that grey color in the boxes does not mean no vocalizations were occurring, but that there was no correlation of call-and-response for those individuals at those times. The main text says “while birds shared many ‘significant interactions’ (non-zero correlation indices, cf. methods for definitions) with other birds in various call types at the beginning of the breeding experiment (many coloured vs. grey squares in Figure 5 to the left), interactions decreased and became more and more individual-specific over the course of the trials (few coloured squares in Figure 5 to the right).” This could be not interactions decreasing, but correlations among many individuals decreasing, and correlations between specific individuals increasing. Can the authors dissociate the decreasing group interactions from increasing pair interactions versus simply less vocalizing in the colony?

Figure 6 on number of call type interactions is the most convincing in this regard and the most convincing figure of the paper, but it still does not separate out amount of vocalizations vs. level of correlations.

Although the authors said that they did not include hormone measures of correlation with call behavior, due to possible temporal differences in behavior-hormone interactions, I think they should still show the negative data as a supplement. It is probably of a real biological significance, as singing in birds is known to be associated with rapid and sustained hormone changes and the authors were able to find hormone changes over the course of days. The authors should also note in the paper that it could be possible that the hormone changes don't correlate with the calls even on a short time scale, but could with singing, which they could measure in a follow up study. If they do not include such data, then the hormone results appears as very tangential in the paper.

I understand that the arc calls were discarded from the analyses, due to them having acoustic structure in between other calls. But it would be nice to include a spectrogram example or two of these calls in Figure 2.

Can the authors include the sample sizes in the figure legends? Were samples run in duplicate or triplicates for each individual in the hormone radioimmunoassays, in order to normalize possible technical variability?

In the Methods, in the subsection “Hormone analyses”, it is surprising to read about an unexpected result that in 77 of 95 samples estrogen (E2) could not be detected. Perhaps there was not a large enough blood sample from each individual? It is surprising that DHT could be measured but not E2. This requires some explanation.

---

## [Author Response]

Reviewer 1:

*Individual-level changes*:

*The paper addresses individual-level changes in vocal output using telemetric microphones to isolate and track vocalizations from individually identified birds. These conclusions, however, are based on the undefended supposition that the microphones mounted on each bird record only vocalizations generated by that bird alone. However, Anisimov et al. (2014;*
*doi:10.1038/nmeth.3114**), utilized telemetric microphone recordings in addition to accelerometer recordings mounted on individual birds and find that the accelerometer, not the microphone, reliably isolated vocalizations from individual birds. Steps need to be taken to defend the premise that the individual-level changes are truly generated by one bird and do not reflect simple changes in proximity to other birds, which are likely to change throughout the course of the experiment. This is an important point as this is what gives the paper its novelty. Providing a histogram of average syllable power recorded from each microphone would allow a clearer rejection of the non-self-generated signal. These histograms would ideally reflect a clear bimodal distribution and would allow the authors to more clearly assert their claims*.

Our methodological approach has been described and used before by Ter Maat et al. (2014), which we cite, but we agree that the information on individual recordings is crucial and deserves more attention in our manuscript. In addition to some changes in the text (Methods subsection “Vocalisation recordings”), we added a figure depicting how the backpacks and microphones were fixed on the birds (Figure 1), and note in Figure 7 that frequencies of on-bird recordings show higher power in the lower frequencies.

As in Ter Maat (2014), we fixed the microphones on the birds so they faced inwards (towards the body), instead of outwards. In this way, the microphone was attached as closely as possible to the sound source and directed towards it. Through the backpack, external sounds were dampened, which reduced the danger of capturing calls of individuals other than the bearer of the backpacks. [3] positioned the microphones “in such a way to place the microphone as close as possible to the beak”, but applied in 90 degrees rotation to our approach – that is, facing outwards. As omnidirectional microphones were used, the recording of external sounds becomes unavoidable, and we believe that this difference in positioning of the microphones has an immense effect that explains the need for additional accelerometers.

In addition, Ter Maat (2014) show a distinction between different sound sources (Figures S4 and S11), and state that: “Normally, since the calls are very soft, there was no discernable sound visible that could be attributed to other animals in the aviary or soundbox. A further check of inadvertently recorded vocalizations from an animal other than the focal individual is provided by the fact that in such a case the vocalizations occurred simultaneously in different channels, which was easily determined. However this occurred rarely. Further, the frequency content of the backpack microphone recording was biased to low frequencies, whereas external signal leaks were characterized by a lack of these.” In our study, too, in the (rare) cases that non-focal birds’ calls had been recorded, they were easily recognised due to frequency properties, and excluded during the semi-automated clustering process. We added this information to the manuscript (in the Methods subsection “Sound analyses”).

*Control for experimental disturbances*:

*The authors present novel insights into vocal communication in the more ethologically relevant context of social housing. It would be useful to control for the effect of backpacks or other disturbances (such as nest checks and blood draws) on vocalizations. A useful control for the former would be to record vocalizations with a house microphone from birds with and without backpacks.*
[3]
*show a transient 2-week suppression in vocalizations upon fitting the backpacks*.

Nest checks and blood sampling are common methods in behavioural biology. However, they might, of course, affect immediate behaviours of the study subjects. Although a possible bias was expected to be similar for all individuals due to equal treatment (which would therefore not alter the outcome of our dataset), we aimed to minimise such effects in the following way, and emphasised this information in the manuscript.

Nest checks were carried out during early afternoon hours when our birds show low activity levels. During this time, we paused the sound recordings and did not perform any behavioural observations. In addition, although sound was recorded almost for the entire day, we analysed morning hours, thus excluding recordings from analyses that had been made after nest checks (see our first response above). To reduce a possible effect of blood sampling on the vocal behaviour of the birds, we went through our data again, and excluded sound recordings that had been made on the same day, after blood sampling (2 mornings). All plots, statistics and sample sizes were adjusted.

We agree that the backpacks have a transient impact on the birds’ behaviour, and are currently preparing to address this in a different, not yet published study (Adreani, D’Amelio, Gill, Ter Maat, Gahr). In this study, we investigate the effects of newly mounted backpacks and also of associated handling (battery exchange) on isolated zebra finch males (n = 7). It shows that with the 0.6g-telemetry equipment used by Ter Maat et al. (2014), birds were able to fly immediately after backpack application, and the amount of calls and movement were no different from pre-backpack values after the fourth day of backpack application.

In the present study, a 1g-backpack was used, and according to our previous observations, one week was sufficient to ensure birds had fully habituated to the backpacks and recovered from any adverse effects. We are aware that the [3] paper reported a longer recovery phase, but we would like to point out that in their case, birds were equipped with a logger of approximately 3g, i.e. the three-fold of the backpack weight used in our study, adding ca. 19% of an average zebra finch body mass (average 15.7g in our colony). Our backpacks added about 6% to the average zebra finch body mass.

Reviewer 2:

*The Methods would benefit greatly from a visual timeline for readers to fully and quickly appreciate the timing and sequence of cage transfers, blood samples, recordings, etc. At present, the methods for the housing section of the study design are confusing. For example, one interpretation in the subsection “Study design, housing and nest material” is that pairs were separated after they formed and that the animals were held in single-sex aviaries to get used to the telemetric backpacks. This is unlikely, since the separation of pair mates is a significant stressor. Alternatively, if the authors intended information on the study set-up to appear prior to the information on study group background, that would be in keeping with a logical, chronological order of the study. Similarly, it is not until later in the section that we learn that the acoustic recordings were not continuous and were restricted to periods of the day. More detail about the recording time intervals is essential, since circadian factors can influence vocalizations in this and other songbird species*.

We appreciate the suggestion of a timeline, which we now added to the manuscript as Figure 8. We also restructured the Methods subsection “Study design, housing and nest material” to make the succession of events clearer. Of course, birds were not paired and then separated again, but were held in large same-sex aviaries before the trials and in smaller same-sex aviaries one week after backpack application, until they were joined in the mixed-sex aviary.

We also emphasized in the manuscript that sound was recorded during the largest part of the day, but only a subset of these recordings were analysed (see the Methods subsections “Study design and housing” and “Vocalisation recordings” as well as Figure 8).

Circadian factors: sound recordings were paused during the birds’ early afternoon hours when our birds show low activity levels (see the Methods subsection “Behavioural observations, nest checks and blood sampling”), so we could carry out nest checks and animal keeping activities (please see above the response to the second concern raised by Reviewer 1). To qualify for analysis, sound files had to be free of external disturbances, i.e. they were recorded before bird husbandry activities. Thus, they started within the first 2 hours of lights on and continued for up to four hours, i.e. they ended before the birds’ midday (the inactive time of the day). We explained this in detail in the Methods subsection “Sound analyses”).

*It is not clear whether the external microphone was used as a reference to gauge the quality of the on-bird microphone recordings. This is important because the harmonic structure of vocalizations is much more apparent up to 8 kHz (*Figures 3 and 8*) in the sonograms from the external microphone, whereas by contrast the sonograms from the on-bird microphones show substantial frequency losses above 2 kHz. This indicates that there is a significant frequency filtering by the on-bird microphones and/or potential filtering by the placement of the microphone on the back of the bird (i.e. muffling by feathers/body). This is especially stark when comparing these recordings to Zann's original 1996 work on zebra finch calls. Zann's work clearly shows that calls have significant acoustical energy up to 8-9 kHz that may carry information about call identity/category. This therefore bears significantly on the experimenter/automated assignment of calls recorded by the on-bird microphones to different call categories*.

External microphones were installed for control purposes, to maintain an overview over the group recordings and vocalisations. No recordings from external microphones were included in the analyses. The spectrogram was added in the manuscript (now Figure 7) to give a visual example of the specificity of recordings compared between external (group) and individual microphones.

Indeed, recordings from individual on-bird microphones contain more power in the lower frequencies than external recordings do. This effect was not only expected, but we actually used it to disentangle individuals, in the (rare) case that non-focal birds’ calls had been recorded (see the first response to Reviewer 1). As described in Ter Maat et al. (2014): “wireless microphones show more power in the lower frequencies as compared with external microphones since they record the near field*.”* We added a note on this important point in Figure 7, and included it in the Methods subsections “Vocalisation recordings” and “Sound analyses”.

In addition, this effect was the same for all microphones, as the distance towards the microphones was the same for all birds (directly on the back). Therefore, this allowed us to use the same classification paradigm for different birds.

Thirdly, a large part of information is contained in lower frequency bands, e.g. the fundamental frequency (one of the most commonly used parameters in bioacoustics), and most important spectral parameters used for classification, such as duration or frequency modulation, can still be used to identify call types. Therefore, our backpack recordings are comparable with Zann’s field recordings, and allow the same call classification.

We do, however, agree that parts of the information contained in higher frequencies get lost using this technique, and do not allow the use of such recordings e.g. as stimulus calls for playback studies. However, the main goal of our study was to get an overview over types and timing of calls, for which the information content of the recordings was easily sufficient.

*The A/D board digitization rate (44 kHz) is good for the quality of the sound recorded, but in this case equally important is the transmission frequency of the on-bird radios. This needs to be provided, although it is likely to be in the megahertz range (the manufacturer's website, sparrow system, is currently unavailable). For example, if the on-bird radios are transmitting at low frequencies (i.e. under 22 kHz), this would reduce the quality of the recordings substantially, and may explain some of the frequency filtering noted above*.

Transmission frequency was between 375–380 mHz. We included this in the manuscript (Methods subsection “Vocalisation recordings”). Lower frequencies from on-bird microphone recordings compared to external microphone recordings are explained in the second response to Reviewer 2.

The extraction efficiency for the plasma steroid extraction should be provided.

Steroid extraction efficiency (mean percentage ± standard deviation = SD of recoveries) was included (Methods subsection “Reproductive stages”): Testo: 70.9 ± 2.7, DHT: 73.7 ±4.3, E2: 59.9 ±4.0, P4: 55.9 ±11.0.

*One of the expectations I had going into the manuscript was that the authors had the opportunity to characterize the behavioral ‘predictors’ of individual call types. That is, they are conducting behavioral observations while also monitoring the vocalizations of individuals. If calls are acting as signals of behavioral intent or responsiveness, the results in this study could provide a much deeper look at vocal and social interactions than what is thus far reported in the study and what is known in this species*.

Our study is not (and did not intend to raise expectations of) a “classical” vocal repertoire study that correlates direct observations of behaviour or contexts 1:1 with vocalisations, as we would require much more fine-tuned behavioural observations and categories to match the fine temporal resolution of rapidly changing calls (e.g. automatic video tracking system). We hope the new version of the manuscript, especially the new Abstract and Introduction, should make this clearer.

Defining the repertoire in more detail in a way that calls could be used to predict (or follow?) certain behaviours could thus be a very interesting follow-up study that would greatly profit from using acoustic telemetry methods.

Reviewer 3:

*The study has challenges that I think need to be fixed. The biggest are that: 1) it under-sells the findings on call interactions in the abstract and beginning of the paper, 2) it over-sells being the first to use such microphone telemetry to record individual vocal interactions in songbirds, and 3) although some of the statistical analyses is based on the latest tools, it not presented clearly enough and some of the findings don't seem to be different as the text indicates. I highlight examples of these three points below*.

We thank the referee for these critical evaluations and reply to them in detail in the following paragraphs. Here, we would just like to point out that in our figures y-axes were kept the same for subplots whenever possible, to make values easier to compare (e.g. the amount of vocalisations in different call types). This results in a lower resolution of some subplots, and as a consequence differences might appear less strong at first glance.

*1) The result mentioned in the Abstract is too non-specific, to the point that it does not say much. I think that several of these sentences should be substituted with ones that are more specific about what happened to the calls in individual pairs of animals, and then a more general conclusion statement about the overall conclusion of the paper – could be that use of this technology allowed them to identify previously unappreciated complex vocal communication interactions between a population of freely behaving birds. I think the authors should begin the Results section of the paper with some more of the exciting findings. Put the hormone findings last, and keep the order of everything else the same*.

We rewrote most of the Abstract and restructured and rewrote parts of the Introduction to make them more to-the-point and with a stronger focus on vocal interactions, and stating more explicitly what happened with vocal behaviour when reproductive stages changed.

To make the manuscript more streamlined, we also moved most of the hormones section (Results and Discussion) to the appendix (Appendix 1), where they were explained and discussed in more detail without diverting the focus of the manuscript. To start off the results with something “exciting”, we also added an overview at the beginning of the Results section.

*2) The authors write several long sentences about how their approach is the first, which almost any long sentence with a lot of qualifiers would be the first of its kind. However, one of the papers they cited (Anisomov et al., 2014 in Nature Methods) is the first real study that used such a telemetry approach to record vocal communication interactions between freely behavior songbirds. The current Gill et al. paper is the second that I am aware of. It looks like both studies were started independently. Nevertheless, the authors should acknowledge the similar approach used in Anisomov and compare and contrast with their approach, at least in the Methods section of the Gill et al. paper. The greater advance that the Gill et al. makes is that they performed and learned a lot more about the biology and behavior of the animals than did Anisomov et al. For that reason too, I think this paper is making an advance*.

We restructured and rewrote most of the Discussion section to decrease the focus on the technological aspect and to shift it more towards the biological knowledge we gained. Also, the according paragraph in the Discussion (“new approach, new results”) was moved to the front of the Discussion, in order to end the discussion with the most important thoughts (calls and interactions and their function in groups).

We now explain the advantages of our study in the Methods section, but do not directly compare the two studies (see Methods subsection “Vocalisation recordings”). Although we did not include the following as explicitly in our manuscript, we would like to point out that what might seem as minor technical differences between our and the Anisimov et al. (2015) study, can have severe implications for the study species and thus on the levels of conclusions that can be drawn (please also see the response to the first concern raised by Reviewer 1). We would like to let readers draw their own conclusions on this, because we did not want to be offensive. If the reviewers and the editor consider it important, we could include a section in the Methods that would direct compare the two studies, similar to the following:

First of all, the backpacks used by Anisimov (2014) weighed 3g, i.e. three times as the ones used in our study, and birds showed a prolonged habituation period. Secondly, birds were handled every morning before the beginning of the sound recordings to exchange logger batteries (3), which was reduced to once a week in our study. Thirdly, to synchronise loggers (clock drift), the backpacks used by [3] contained infrared sensors that required visual contact with the diode at all times. Apart from adding more weight to the backpacks, this required that the environment in which birds were housed was unstructured and limited to two dimensions. Thus, birds were held in small enclosures (60 x 60 x 50 cm), with perches on the ground (3), which, despite what authors claim, we believe does not give the birds the opportunity to fly anywhere, and strains the notion of “freely behaving animals”. Fourthly, in addition to the technological differences, bird “groups” consisted either of singly housed pairs, or of a maximum of four males (3). This is why we speak of the “first time” that mixed-sex bird groups were investigated – in a setting that would allow an investigation of more “normal”, species-relevant behaviour.

*3) The box plots in*
Figures 1 and 3
*and*
Figure 4—figure supplement 1
*have a lot of information, which is not explained well enough. Most readers will look to see if there is overlap in the 95% CI boxes, and then assume that there are nearly no differences among groups. The authors should point the reader to the means being different (thin horizontal black line) in the legend. It is not clear to me what the vertical Bayesian CI gain is for this analysis? Why did the authors perform this statistic and add it to the box plots? Would the comparisons be significant without doing the Bayesian analyses and using standard repeated measures ANOVA? The letters a, b, and c in the box plots for statistical differences need a better explanation of what group is each letter comparing. The authors should also explain on first use in the results what does F, and R*^*2*^_*marginal*_
*vs. R*^*2*^_*conditional*_
*mean. This is not a typical statistic used in papers, and they cite it as something new in 2013 in the Methods, but without any explanation*.

We used boxplots to represent the distribution of the real data in a summarised way. The red or blue points and vertical lines added to the boxplots represent the results from the according statistical evaluation (Bayesian estimates and credible intervals). These aspects are explained in more detail below. We added a better description of the Bayesian statistical tools that we used in the Methods subsection “Statistical analyses”, including a better explanation of F and the two R square values which we refer to throughout the manuscript. Note that F is one of the most common test statistics reported in papers, and in our view, does not require a more thorough explanation.

Bayesian statistics have been available for a while now, but only recently, since computers have become efficiently fast (large computational power required in simulations), did they become accessible to a wider range of users. Even in various fields of biology, they have become appreciated, because they can provide solutions to problems previously difficult to address using frequentist analyses. We do not intend to engage in the ongoing and sometimes fierce discussions about which of these methods is “better”. We decided to use Bayesian statistics and an information technological approach because, among other things, drawing inference from mixed models is not possible in the frequentist approach (10). Instead of formulating and rejecting non-meaningful null hypotheses, the Bayesian approach is used to investigate differences in effect sizes which can be evaluated directly and intuitively between all groups, by comparing posterior (i.e. estimated from the model) means and credible intervals. Opposed to this, “standard repeated measures ANOVA” would require post-hoc analyses for all groups, and multiple testing decreases statistical power.

To keep the relationship between the model and the actual values accessible for the reader, it is common practice to provide both simulated and actual values. In our view, the most efficient way of doing so if only few categories are compared, is visually, by plotting the raw data along with the means and credible intervals coming from the model, instead of providing tables with numbers. We had plotted the raw data points at first, but decided, for the sake of clarity, to use boxplots, as they are a commonly used and a very efficient way of data representation. We now added better descriptions of what each line and symbol means to the legend of Figure 4 (and Figure 6), which we refer to in the other plots where appropriate. We also changed the abbreviation “CI” into “CrI” to prevent readers from automatically thinking of “confidence intervals” but rather reminding them that we are referring to “credible intervals”. For the sake of clarity, in the case of Figure 4—figure supplement 1 (which should only give an overview over the proportion of call types in the repertoire over more detailed breeding stages), we now provide boxplots with Bayesian CrI and estimates, but, instead of indicating differences between groups via different letters, added a table with values (Appendix 2) for interested readers.

*Specific important comments*:

*The authors use the word “natural” throughout the paper, and in the Introduction imply that they will study natural calling behavior, which they say little has been done before on. I am not aware of unnatural experiments. In addition, the authors perform experiments indoor, not in the natural wild environment, like most other investigators have. So I would not exactly consider this a natural experiment. This is another over-sold point, so much so that they attack some of the prior literature unnecessarily*.

We very much agree that this is not a field study. We had used the term “natural” four times in this version of our manuscript (plus a few times for the “natural logarithm”). The context in which we had used this was that we investigated “naturally”, i.e. spontaneously occurring vocalisation interactions between birds in direct visual, acoustic and physical contact, as opposed to e.g. playback studies in which a bird is giving calls in response to a timed succession of recorded bird vocalisations.

In our view, we had formulated this carefully, never including the term “natural experiment”, as our study neither takes place in nature, nor is, strictly speaking, an experiment, but rather a descriptive study. However, we went through the manuscript once more, to make sure that this impression would not arise. We therefore changed the wording in the Introduction and in the Discussion, where the term might have been ambiguous, and refer to “naturally occurring vocal interactions”, “reduced social contexts or impoverished environments”, and now refer to “relevant context”. Other sentences were removed from the manuscript due to major structural changes.

We did not intend to attack previous studies that had worked with birds in isolation, as this can be a useful tool for answering specific questions. However, housing a highly social, group-living bird like the zebra finch in social isolation for many days constitutes a “strongly reduced social environment” which is very likely to affect the focal birds’ vocal behaviour, and hardly supports functional investigations of vocalisations.

*I like*
Figure 5*, which is a novel view to me about displaying behavioral interactions. But it can use more explanation in the Methods or in the legend*. *Does the position of the data dots (turquoise to red) within the small tiny grey squares mean anything?*

Figure 5 is a visual representation of the raw information (from PSTHs) stored in contingency tables, and in our view, a very efficient way of demonstrating the different levels of communication. It does, however, require the reader’s full attention.

PSTHs are commonly used in sciences comparing onsets of events (e.g. neuroscience) and correlation values are calculated relative to baseline levels in each dyad (see Methods). They are explained in more detail in a similar study on vocal interactions by Ter Maat et al. (2014).

We improved the descriptions (size and colours of legends) in Figure 5 to make the levels of information more accessible that are encoded in the different levels of the correlation matrices (bird level and call-type level). We hope this gives more explanation and answers what “the position of the data dots (turquoise to red) within the small tiny grey squares mean”. We gave the new version of this figure to students not involved in our study and to non-scientist persons, and are happy to announce they explained it to us correctly. If still considered necessary, we could also add a figure supplement with a step-by-step explanation on how to read and interpret this figure, to make the information more accessible to readers not familiar with this kind of presentation. We would like to leave the decision on whether to include such a figure supplement or not up to the editor.

*The authors should emphasize that grey color in the boxes does not mean no vocalizations were occurring, but that there was no correlation of call-and-response for those individuals at those times*.

We agree that it might be helpful to emphasize this, and implemented this in the manuscript (in the Introduction and legend of Figure 5).

*The main text says “while birds shared many ‘significant interactions’ (non-zero correlation indices, cf. methods for definitions) with other birds in various call types at the beginning of the breeding experiment (many coloured vs. grey squares in*
Figure 5
*to the left), interactions decreased and became more and more individual-specific over the course of the trials (few coloured squares in*
Figure 5
*to the right).” This could be not interactions decreasing, but correlations among many individuals decreasing, and correlations between specific individuals increasing*.

We disagree with the reviewer, on this point. As mentioned before, Figure 5 depicts the “raw” interaction data coming from PSTHs of all possible combinations, and only those that reach the significance criterion pop up in colour. In the new version of the manuscript we have ensured that the reader can easily understand what the position and colour of each data point means and thus he or she will be able to see who communicated with whom when and how intensively, and will therefore be able to disentangle “interactions decreasing” from “correlations among many individuals decreasing, and correlations between specific individuals increasing”. We hope that the newly added information mentioned above will help to clarify this.

Can the authors dissociate the decreasing group interactions from increasing pair interactions versus simply less vocalizing in the colony?

To some extent, these two aspects of vocal communication are correlated. If birds do not vocalise at all (which never occurred), they could, of course, not exhibit any vocal interactions. In addition, if birds interact vocally with fewer individuals, it is likely that they vocalise less in total.

To make the information about the amount of vocalisations and interactions in relation to the specificity of vocal interactions within pairs at different breeding stages available at first glance, we added another figure (Figure 5—figure supplement 1, mentioned in the first paragraph of the Results and the subsection “Vocal interactions, reproductive stages and successful egg-laying”) showing the vocal activity and the intra-pair and extra-pair vocal interactions in the same graph. Note that although vocal activity and the number of overall interactions decrease slightly, the specificity of within-pair interactions increases over breeding stages.

Figure 6
*on number of call type interactions is the most convincing in this regard and the most convincing figure of the paper, but it still does not separate out amount of vocalizations vs. level of correlations*.

As mentioned above, PSTHs are used to calculate correlation values between occurrences, relative to a baseline. In addition, we added a summarising figure (Figure 5—figure supplement 1) to answer this question.

*Although the authors said that they did not include hormone measures of correlation with call behavior, due to possible temporal differences in behavior-hormone interactions, I think they should still show the negative data as a supplement. It is probably of a real biological significance, as singing in birds is known to be associated with rapid and sustained hormone changes and the authors were able to find hormone changes over the course of days. The authors should also note in the paper that it could be possible that the hormone changes don't correlate with the calls even on a short time scale, but could with singing, which they could measure in a follow up study. If they do not include such data, then the hormone results appears as very tangential in the paper*.

Hormone samples were collected at one time point before and at two time points during the trials, to give a large-scale overview over the birds’ physiological status while the birds went through changes in life-history stages. As stated in our manuscript (in the Discussion subsection “Opportunistic breeding and vocalisations “and also in the third paragraph of Appendix 1), it is not yet possible to draw a direct comparison between hormones and calls due to extreme differences in the temporal resolution of sampling methods available to date. Therefore, we did not design the study in a way to address such correlations, and hormones were not systematically sampled on the same day as vocalisations – which makes sample sizes very low, and a direct comparison rather difficult. A different study design would be needed to address these kinds of questions, e.g. by manipulating hormones of the individual birds in a physiologically relevant range.

We now moved most of the hormone section to Appendix 1, as hormones were not the main focus of this study, and we realise the previous version might have made this impression before. We added some speculations, also in reference to previous literature, on hormones and vocalisations to the appendix (in the third paragraph of Appendix 1). In the main text, we now only briefly refer to hormones (e.g. Introduction, Discussion subsection “Opportunistic breeding and vocalisations”) and synchronisation (in the same subsection of the Discussion).

*I understand that the arc calls were discarded from the analyses, due to them having acoustic structure in between other calls. But it would be nice to include a spectrogram example or two of these calls in*
Figure 2.

In Figure 3, we show the five call types that we used for further analysis. Adding spectrograms of arc calls might thus make it confusing which call types we used and which ones we didn’t.

Can the authors include the sample sizes in the figure legends? Were samples run in duplicate or triplicates for each individual in the hormone radioimmunoassays, in order to normalize possible technical variability?

Sample sizes were included in figure legends if appropriate. All samples were run in duplicates (see the Methods subsection “Reproductive stages”).

*In the Methods, in the subsection “Hormone analyses”, it is surprising to read about an unexpected result that in 77 of 95 samples estrogen (E2) could not be detected. Perhaps there was not a large enough blood sample from each individual? It is surprising that DHT could be measured but not E2. This requires some explanation*.

We do not find this surprising because, according to our experience, E2 levels are usually very low in zebra finches and most other songbirds. The fact that DHT was detectable simply means that DHT levels were higher than E2 levels because the detection limits are rather similar between the two hormones. A largely increased amount of plasma would most probably have made E2 detectable, but this was not possible in our setup due to animal welfare regulations.